# STREAMING VIDEO UNDERSTANDING AND MULTI-ROUND INTERACTION WITH MEMORY-ENHANCED KNOWLEDGE

**Haomiao Xiong**[1]*, **Zongxin Yang**[2]*, **Jiazuo Yu**[1], **Yunzhi Zhuge**[1]†, **Lu Zhang**[1], **Jiawen Zhu**[1], **Huchuan Lu**[1]

[1]Dalian University of Technology, [2]Harvard University

## ABSTRACT

Recent advances in Large Language Models (LLMs) have enabled the development of Video-LLMs, advancing multimodal learning by bridging video data with language tasks. However, current video understanding models struggle with processing long video sequences, supporting multi-turn dialogues, and adapting to real-world dynamic scenarios. To address these issues, we propose STREAMCHAT, a training-free framework for streaming video reasoning and conversational interaction. STREAMCHAT leverages a novel hierarchical memory system to efficiently process and compress video features over extended sequences, enabling real-time, multi-turn dialogue. Our framework incorporates a parallel system scheduling strategy that enhances processing speed and reduces latency, ensuring robust performance in real-world applications. Furthermore, we introduce STREAMBENCH, a versatile benchmark that evaluates streaming video understanding across diverse media types and interactive scenarios, including multi-turn interactions and complex reasoning tasks. Extensive evaluations on STREAMBENCH and other public benchmarks demonstrate that STREAMCHAT significantly outperforms existing state-of-the-art models in terms of accuracy and response times, confirming its effectiveness for streaming video understanding. Code is available at StreamChat.

## 1 INTRODUCTION

Recent advancements in Large Language Models (LLMs) [1–3] have led to the development of Video-LLMs [4–9], which aim to interpret visual scenes, actions, and narratives. These models represent significant progress in multimodal learning by bridging video data and language-based tasks, with applications spanning from content analysis to human-robot interaction [9].

Despite these advancements, current offline models primarily process videos as static clips and rely on single-turn dialogues, incorporating visual information through mechanisms like projection layers [4, 8] or cross-attention structures [2]. However, these models encounter computational bottlenecks when handling extended video sequences, often struggling to compress

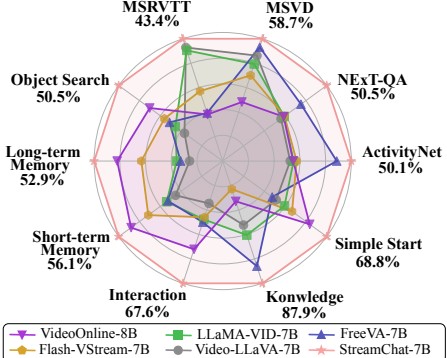

Figure 1: **Performance comparison** between STREAMCHAT and previous Video-LLMs.

lengthy video features within limited memory resources [10]. Additionally, their inability to support multi-turn dialogues reduces adaptability for interactive scenarios, and key information may be lost due to insufficient video sampling methods (*cf.* Fig. 2(a)).

To address these issues, online models [10, 11] have emerged. They utilize memory-based approaches and temporally aligned instruction-tuning to process long videos and enable multi-round interactions (*cf.* Fig. 2(b)). While these models allow dynamic user interactions, they still face challenges

---

*Equal contribution.

†Corresponding author (zgyz@dlut.edu.cn).

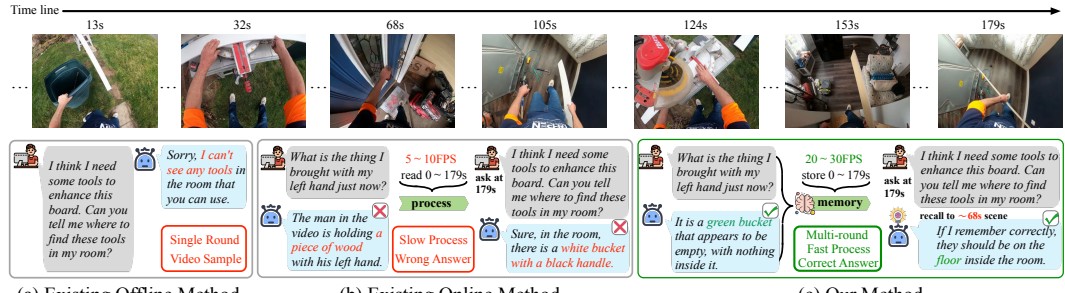

Figure 2: **The comparisons between StreamChat and other methods** (§1). Offline methods process entire videos, leading to information loss and limited to a single interaction. Previous online methods [10, 11] enable multi-round interactions but still suffer from slow processing and answer correctly. The proposed method achieves real-time video processing, improving the efficiency and accuracy with memory support.

maintaining rapid processing speeds and consistent performance across unfamiliar scenarios—critical factors in real-time applications like robotic navigation and human-robot collaboration.

To overcome these limitations, we propose STREAMCHAT, a training-free framework for streaming video understanding that offers three key innovations: **(i) Training-free adaptability**, allowing it to efficiently process videos of various types and lengths without resource-heavy training. This makes STREAMCHAT suitable for both online and offline video processing while maintaining stable performance across diverse scenarios. **(ii) Hierarchical memory storage**, which manages and compresses video information over long sequences. It integrates short-term memory for tracking ongoing events, long-term memory for retaining past events in compressed form, and dialogue memory to maintain conversational history, ensuring continuous and coherent dialogue understanding. **(iii) Optimized system scheduling**, which improves model inference efficiency by processing tasks in parallel across three threads: the *selective frame stacking* thread identifies and removes redundant frames, the *memory formation* thread updates and refines memory information, and the *contextual summarization* thread handles user requests and generates responses in real-time.

We evaluate STREAMCHAT on existing benchmarks [12–16] and identify their two major shortcomings: **(i) Short and monotonous video content**, which fails to capture the complexity of real-world streaming media, and **(ii) Simplistic, single-round questions**, which do not test the model's ability to engage in multi-turn dialogue or complex reasoning.

To address these deficiencies, we introduce STREAMBENCH, a comprehensive benchmark designed for streaming video understanding. It includes a diverse array of video content such as egocentric videos, web videos, and movie scenes, paired with text annotations that simulate multi-round interactions. *In terms of video selection*, we perform rigorous manual curation from large datasets to ensure both high-quality content and a broad range of categories. *In terms of questions*, we design six distinct types of queries, probing various dimensions of the model's reasoning abilities, from simple factual retrieval to complex inference. Compared to previous benchmarks, STREAMBENCH not only evaluates the accuracy of model responses but also incorporates latency metrics, which are essential for assessing performance in real-time applications. This comprehensive evaluation framework offers a more realistic and reliable measure of model robustness and practical utility.

In summary, our key contributions are as follows:

- We propose STREAMCHAT, a training-free method for streaming video understanding. Its novel *hierarchical memory storage* and *system scheduling* strategy enables robust memory management, real-time video processing, and multi-round interaction capabilities. These features ensure precise and efficient response generation, catering to the dynamic nature of video contexts.

- We introduce STREAMBENCH, the first comprehensive benchmark to evaluate streaming video understanding models. This benchmark simulates real-world interactions through multi-turn dialogues and diverse question formats, offering a detailed assessment of model performance.

- STREAMCHAT sets new benchmarks (*cf.* Fig. 1), delivering a 64.7% accuracy on STREAMBENCH for online settings, which is an 8.3% improvement over the previous best. In offline scenarios, it outperforms the state-of-the-art method by an average of 2.5% across four public benchmarks.

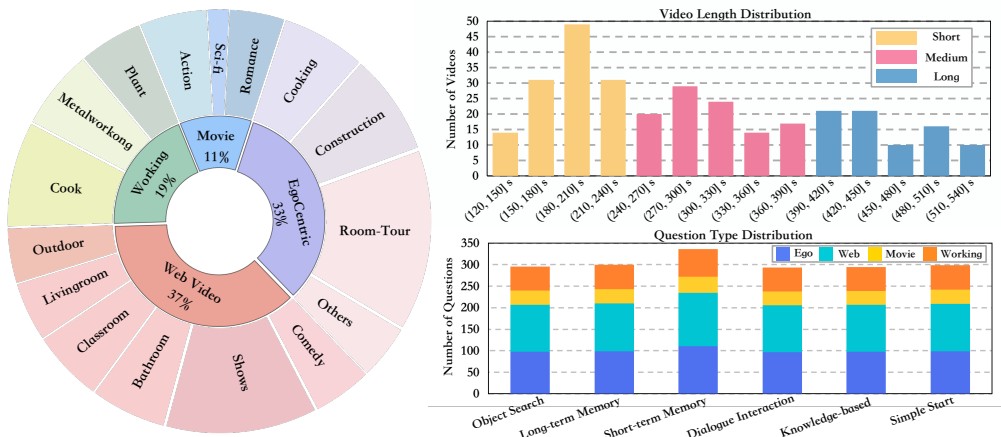

Figure 3: **Benchmark overview** (§2). Our benchmark covers 4 key domains and 16 sub-class video types. These videos exhibit a broader distribution of length, with 6 different types that are evenly distributed.

- In terms of efficiency, STREAMCHAT achieves a processing speed of 32 FPS, marking a sixfold increase over existing methods. Additionally, it maintains text generation latency under 0.9 seconds, showcasing significant advances in interactive video processing.

## 2 COLLECTION AND COMPOSITION OF STREAMBENCH

### 2.1 VIDEO COLLECTION

Previous video understanding benchmarks [12–14, 16–18] primarily focus on offline scenarios, where all video frames and user questions are provided to the model simultaneously for generating answers. To find a more suitable method to assess a model's ability to understand online scenarios, we introduce STREAMBENCH, a benchmark mainly designed to simulate online video scenarios. There are two distinct differences compared with other video understanding benchmarks. **(i) Diverse video curation**: we

Table 1: **Comparisons of different benchmarks**. MRI denotes multi-round interactions.

| Benchmark | MR | Avg | Total | Video | QA |
|---|---|---|---|---|---|
| MSVD [16] | ✗ | 10s | 1.4h | Web | Desc. |
| MSRVTT [12] | ✗ | 15s | 12.5h | Web | Desc. |
| ActivityNet [14] | ✗ | 112s | 25h | Web | Desc. |
| Next-QA [15] | ✗ | 40s | 11h | Web | Temporal |
| MovieChat [7] | ✗ | 213s | 9h | Movie | Movie |
| STREAMBENCH | ✓ | **270s** | **25h** | **Mix** | **Online** |

collect four major domains and sixteen sub-classes of video sources, including egocentric videos, web videos, working videos, and movies as the database of the benchmark. Each type has its unique characteristics and challenges, which can verify the stability and reliability of the model in a wide range of application scenarios. **(ii) Crafted query types**: we design six types of questions to meet the specific needs of online video understanding and ensure that these types of questions appear once in a single video, forming a multi-round dialogue. This section introduces how we collect videos and construct annotations. More details about the diversity and distribution are shown in Fig. 3.

**Data Sources**. In selecting videos for our benchmarks, we prioritized diversity in type and length to maintain high data quality. Our primary sources are the EgoSchema [13] and YouTube-8M [19] datasets. EgoSchema offers a rich array of both indoor and outdoor scenes, providing an extensive range of egocentric perspectives and actions, which aligns perfectly with our experimental needs. From YouTube-8M, which features a comprehensive internet-sourced collection spanning over 4,000 classes, we filtered to procure high-quality web, work-related, and cinematic videos. This diverse selection framework ensures our model is tested against a broad spectrum of real-world scenarios.

**Filtering videos**. It is a crucial step to ensure the quality of the videos used in the benchmark. To achieve this, our data filtering pipeline consists of machine and human selection. Firstly, a multi-modal language model [20] is utilized to classify the original data. The categories of videos are provided by data source, we feed them with the videos to the machine and make it select the category of the video. Secondly, human judgment is required to assess the redundancy: the change of scenes in videos. Some static video content (e.g., ego view of drawing, watching TV) and high-noise data from web videos (e.g., video games, advertisements) are removed according to human judgment. Finally, StreamBench consists of **306 videos** with a total duration of **24.8 hours** and an average of **4.5 minutes** each, offering a comprehensive collection of videos from different categories and lengths.

## 2.2 CONSTRUCTION OF TASKS AND ANNOTATIONS

We have crafted six distinct tasks with annotations to simulate the conversation between the agent and the human. Each task corresponds to a different real-world scenario, ensuring comprehensive coverage of potential communication contexts.

- **Object Search (OS):** Challenges include accurately describing an object's position in a video. The task conditions are that the object must appear for less than 5 seconds and the interval from its appearance to the user's request should exceed 30 seconds, enhancing the difficulty of the search.
- **Long-term Memory Search (LM):** This task assesses the model's memory by requiring recall of events appearing for more than 5 seconds, with a delay exceeding 1 minute from the event's end to the user's query, testing long-term memory retention.
- **Short-term Memory Search (SM):** To simulate the user's interest in recent events, this task sets the interval from event completion to the user's query at less than 20 seconds, evaluating the model's response to recent activities.
- **Conversational Interaction (CI):** Sometimes the answer to a user's current question is closely related to conversation history. Therefore, the model must memorize conversation records and retrieve the most relevant text from the memory as contextual support. This type is designed to simulate multi-turn dialogue scenarios. We set the dialogue information associated with the user's current request to come from any previous conversation, with an interval of more than 2 dialogues.
- **Knowledge-based Question Answering (KG):** This type of question evaluates the model's internal knowledge, which is retained by the base large language models. In this benchmark, we set the questions must be related to the events or objects occurring in the video so that it can simulate scenarios where users have a specific need to understand background or encyclopedic knowledge.
- **Simple Factual (SF):** This type of question focuses on friendly dialogue starting between the user and the model. Therefore, they must be asked within 30s after the beginning of the video. Although the question is simple, the model needs to remember things in the short term to answer correctly.

To ensure the quality of the annotations, we additionally assign different workers to perform human feedback for manual annotation. The human feedback step needs to focus on three parts: (1) check the question formats are correct and diverse, (2) ensure the expressions are clear and consistent with the video, and (3) remove sensitive topics such as those questions related to nationality or politics. These steps of manual annotation and feedback, along with multi-modal large language model assisted video collection, form our semi-automated benchmark construction pipeline (Appen. §A). Finally, STREAMBENCH contains **1.8K** high-quality QA pairs. The distribution of these annotations is shown in Fig. 3. Some examples from the benchmark that offer an intuitive observation of our annotation results and formats are shown Appen. §B.

## 3 STREAMCHAT

Given streaming broadcast video and timestamped questions as input, STREAMCHAT is designed to efficiently perform reasoning and deliver accurate answers across multiple rounds. Building upon LongVA [20] (Appen. §F) as a foundational Video-LLM, our design incorporates two key components: a *hierarchical memory storage* system (§3.1) that leverages long-term, short-term, and dialogue memories to compress and manage extensive video sequences within constrained resources, thereby facilitating effective video-content reasoning; and a *system scheduling* strategy (§3.2) that decouples video feature extraction from memory updates, thereby preventing unbounded buffer growth as the input video frames increase.

Table 2: **Comparisons of recent video-mllms**. Our *streaming* (S.) method with *memory* (M.) achieves processing video in *real-time* (R.) and generates a response with *low latency* (L.).

| Method | S. | M. | R. | L. |
|---|---|---|---|---|
| Video-LLaVA [4] | ✗ | ✗ | ✗ | ✗ |
| MovieChat [7] | ✗ | ✓ | ✗ | ✗ |
| Video-online [11] | ✓ | ✗ | ✗ | ✗ |
| Flash-VStream [10] | ✗ | ✓ | ✗ | ✗ |
| STREAMCHAT | ✓ | ✓ | ✓ | ✓ |

## 3.1 HIERARCHICAL MEMORY STORAGE

STREAMCHAT treats videos as dynamic information repositories, utilizing hierarchical memory to analyze and store the diverse content. This section details two specialized memory structures devised to address the challenges of information storage and retrieval: long-short term memory

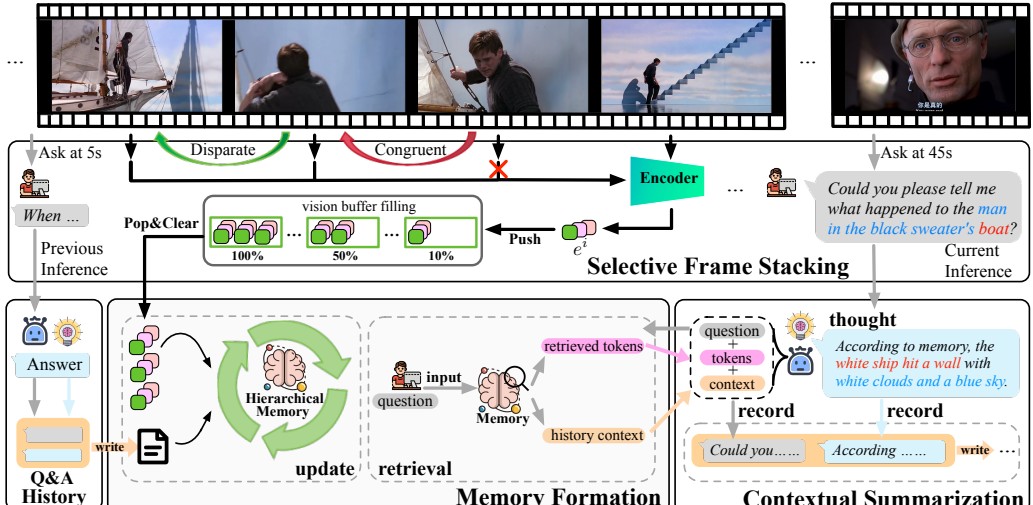

Figure 4: **Overview of StreamChat** (§3), which comprises three main components: **(i) Selective frame stacking**, which prepares vision features for processing, including encoding frames and filling the vision buffer; **(ii) Memory formation**, where vision features are organized into structured memory; **(iii) Contextual summarization**, utilizing hierarchical memory to respond to user queries by providing relevant context.

$M_l \cup M_s = \{l_i\}_{i=0}^{T/L} \cup \{s_i\}_{i=0}^{S}$ and dialogue memory $M_d = \{d_i\}_{i=0}^{D}$. These memories manage visual and conversational data, where $T$ is video duration, $S$ is short memory length, $D$ counts dialogues, and $L$ is the chunk size for long memory. The following sections introduce the functions of the above parameters.

### 3.1.1 LONG-SHORT TERM MEMORY

**Selective Frame Stacking.** To reduce the feature storage overhead caused by redundant frames in videos, we use Lucas-Kanade Optical Flow algorithm [21] in the selective frame stacking module to assist in determining the validity of each video frame $\{F^i \in \mathbb{R}^{H \times W \times 3}\}_{i=0}^{T}$. Specifically, we calculate the motion vector $(u, v)$ between $i$-th frames $F^i$ and the last frame $F^{i-1}$:

$$\begin{bmatrix} u \\ v \end{bmatrix} = \begin{bmatrix} \sum_i I_x(i)^2 & \sum_i I_x(i)I_y(i) \\ \sum_i I_x(i)I_y(i) & \sum_i I_y(i)^2 \end{bmatrix}^{-1} \begin{bmatrix} \sum_i -I_x(i)I_t(i) \\ \sum_i -I_y(i)I_t(i) \end{bmatrix} \tag{1}$$

where $I_x(i), I_y(i), I_t(i)$ represent the partial derivatives of the frame $F^i$ with respect to position $i(x, y)$ and time $t$. We develop the motion vector magnitude $||\theta|| = \sqrt{u^2 + v^2} \in [0, 1]$ to represent the total motion intensity between frames. If $||\theta||$ exceeds the predefined threshold $t \in [0, 1]$, the frame $F^i$ will be encoded into vision embedding $e^i \in \mathbb{R}^{n \times d}$ and pushed into buffer $\mathcal{B}_{\text{vision}}$.

**Short-term Memory.** We intend to design a human-like memory method that simulates the Atkinson-Shiffrin model [22], which emphasizes the role of a short-term storage for maintaining readily accessible, frequently updated information. Specifically, as shown in Fig. 5 (a), we select $N$ vision embeddings from the $\mathcal{B}_{\text{vision}}$ as vision candidates $\mathcal{C}$. Building on the Ebbinghaus Forgetting Curve theory [23], we handle memory updates by randomly selecting $S$ vision embeddings $e^i$ from $\mathcal{C}$ to construct the short-term memory $M_s$:

$$\mathcal{C} = \{\sigma_{N-1} e^{i-(N-1)}, \sigma_{N-2} e^{i-(N-2)}, \dots, \sigma_0 e^i\} \xrightarrow[\text{select}]{\text{random}} M_s = \{s_i \in \mathbb{R}^{n \times d}\}_i^S \tag{2}$$

where $\sigma_i$ is the normalized forgetting probability of $i$-th unit of $\mathcal{C}$, $S$ represents the length.

**Long-term Memory.** The long-term memory simulates the complex and abstract memory of humans [22]. For this reason, we design two forms of information in long-term memory: *text clues*, which is used to store declarative text $t_i$ describing events that occurred over a past period, and *vision memory*, which is used to store compressed visual features $v_i \in \mathbb{R}^{C \times d}$. *Text clues* serve as an index for retrieving relevant information from the long-term memory (introduced in §3.1.3). Our system overcomes the bottleneck of VRAM consumption and the challenges of retrieving memory units $l_i$ by constructing a tree structure as shown in Fig. 5 (a).

The construction of the long-term memory tree can be outlined in the following steps: *Firstly*, the vision buffer is chunked, and each chunk is clustered and assigned a caption:

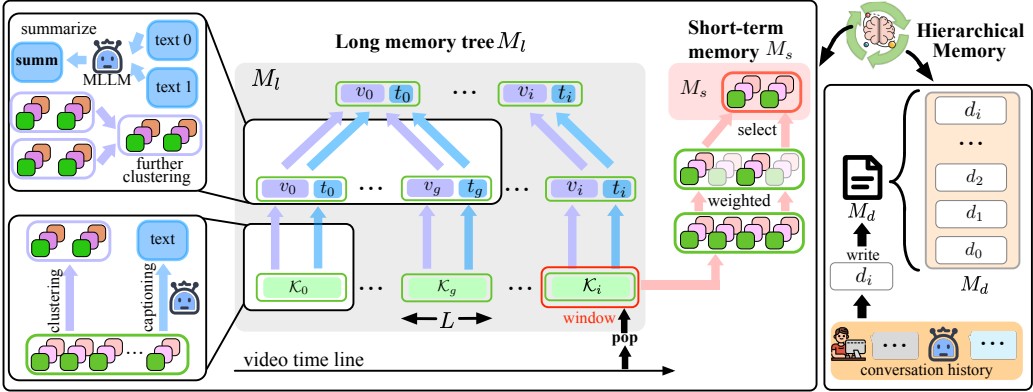

(a) Long-short term Memory       (b) Dialogue Memory

Figure 5: **The hierarchical memory storage** (§3.1). (a) Long-short term memory, where the long memory tree $M_l$ and short-term memory $M_s$ are constructed along the video time line. (b) The dialogue memory $M_d$ is updated after each inference conversation for managing the dialogue histories.

$$\mathcal{B}_{\text{vision}} = \{\mathcal{K}_i\}_{i=0}^{T/L}, \mathcal{K}_i = \{e^i\}_{i=0}^{L}, v_i = f_{\text{k-means}}(\mathcal{K}_i), t_i = p_\theta(x_i|\mathcal{K}_i) \tag{3}$$

where $v_i \in \mathbb{R}^{C \times d}$ is the $i$-th cluster formed from feature chunks $\{\mathcal{K}_i\}_{i=0}^{T/L}$, $C$ is the clustering goals , and $t_i$ represents the $i$-th caption of each chunk. Each chunk $\mathcal{K}_i$ contains $L$ features $e^i \in \mathbb{R}^{n \times d}$ which come from buffer $\mathcal{B}_{\text{vision}}$. *Next*, a clustered feature $v_i$ and a caption $t_i$ together form a long memory unit $l_i$, which also serves as the basis nodes of our tree structure:

$$[l_0, l_1, \ldots, l_{i-1}]_{\text{nodes}} \xleftarrow{\text{push}} l_i = \{v_i, t_i\} \tag{4}$$

Finally, basic nodes are further grouped into higher level nodes $[l_0^1, l_1^1, \ldots, l_k^1]$ in chronological order until a tree structure is formed and all the basic nodes are exhausted :

$$M_l = \{l_i\}_{i=0}^{T/L}, l_k^1 = \{f_{\text{k-means}}(\{v_i\}_{i=0}^g), p_\theta(x_i|\{t_i\}_{i=0}^g)\} \tag{5}$$

### 3.1.2 DIALOGUE MEMORY

In our approach, each round of question $\{Q_i\}_{i=0}^D$ and answer $\{A_i\}_{i=0}^D$ is viewed as a memory fragment, which is pre-encoded by the encoder model $E(\cdot)$ into a contextual representation $d_i$. Thus, the entire dialogue history $M_d$ is pre-encoded as shown in the following formula: $M_d = \{d_0, d_1, \ldots, d_{i-1}\} \xleftarrow{\text{push}} d_i = E(<Q_i, A_i>)$ where the length of $M_d$ is equal to the conversation number $D$, and we select MiniLM-L6 [24] as our encoder model.

### 3.1.3 RETRIEVAL

When a user question $Q_i$ comes in, the memory system will search for the most relevant knowledge as supplementation by the retrieval algorithm. In long memory tree $M_l$, the $Q_i$ and text clue units $\{t_i\}_{i=0}^{T/L}$ are encoded by the tokenizer and the embedding layer of LLM. Based on the cosine similarity between encoded $Q_i$ and text clue $t_i$, the memory system will search for the *retrieved tokens* $M_s \cup \{v_r \in \mathbb{R}^{C \times d}\}_{r=0}^{\mathcal{L}}$ where $\mathcal{L}$ represent the layer number of $M_l$. In dialogue memory $M_d$, the user requests $Q_i$ is encoded by $E(\cdot)$ as a query to search for the *context* $<Q_{\text{retrieved}}, A_{\text{retrieved}}>$ based on the FAISS [25] index. More details about our retrieval algorithm are shown in Appen. §C.

### 3.2 SYSTEM SCHEDULING

As shown in Fig. 4, our method includes three different parts: selective frame stacking, memory formation and contextual summarization. These components are operated as independent threads to optimize inference speed and minimize latency. System scheduling is crucial as it enables concurrent execution of these threads without interference, significantly enhancing processing speed.

Specifically, the **(i)** *slective frame stacking thread* actively populates the vision buffers $\mathcal{B}_{\text{vision}}$ with features $e^i$. Once full, these features are cleared from the buffer and passed to the **(ii)** *memory formation thread*, which updates the memory structures by building the long-term memory tree $M_l$ and refreshing the short-term memory $M_s$. Concurrently, previous dialogue records ($<Q_{i-1}, A_{i-1}>$) are stored in the dialogue memory $M_d$. Upon receiving a new query $Q_i$, the **(iii)** *contextual summarization thread* retrieves relevant information from the hierarchical memory to provide timely responses. This architecture supports sub-second latency (<0.9s) and video processing up to 32 FPS.

# 4 EXPERIMENTS

## 4.1 EXPERIMENTAL SETUP

**Memory Configurations**. To adapt the model to various application scenarios, we configure three versions with different memory settings: Base, Fast, and Slow. These variants adjust key memory parameters, including threshold ($t$), chunk length ($L$), group size ($g$), and clustering goals ($C$), as summarized in Tab. 3. The Fast model is optimized for rapid video processing, while the Slow model prioritizes accuracy in responses. The Base model balances processing speed and accuracy.

Table 3: **Memory configurations** for three models.

| Version | $t$ | $L$ | $g$ | $C$ |
|---------|------|-----|-----|-----|
| Slow | 0.13 | 35 | 15 | 5 |
| Base | 0.35 | 25 | 10 | 5 |
| Fast | 0.58 | 30 | 15 | 5 |

**Evaluation Metrics.** We evaluate semantic similarity in single conversations using the LLaMA-3 model [3], which assigns a semantic correctness score (Sco.) ranging from $[0, 5]$, where higher scores reflect responses that more closely align with the expected answers. For assessing coherence in multi-turn dialogues, we compute score fluctuations across turns; smaller fluctuations (Coh.) indicate a smoother dialogue experience. Additionally, we measure request processing delay (RPD), defined as the time (in seconds) from user request submission to the start of response generation. A smaller RPD signifies lower latency, resulting in reduced wait times for users. Appen. §D offers more details.

**Implementation Details.** We utilize CLIP-L-P14 [26] as the vision encoder and we set the number of selected memory units $S$ to 5 and candidate length $C$ to 20. Experiments were conducted on two NVIDIA Tesla A800 GPUs with 80GB of memory each (more details in Appen. §F ). We benchmark our model against state-of-the-art methods, including Video-LLaVA [4], LLaMA-VID [2] and etc.

## 4.2 COMPARISON WITH STATE-OF-THE-ART METHODS

**Online Scenarios.** As shown in Tab. 4, our models demonstrate significant improvements over the previous best method, Video-online [11].

- Slow: Achieves an **8.3%** higher accuracy and a **0.37** higher score than Video-online.

- Fast: Processes video at 32 FPS, making it much faster than all previous streaming methods, while still improving accuracy by **5.3%** and scoring **0.17** higher than [11].

- Base: Reaches 63.8% Acc. and 3.42 score.

- Best model: Surpasses [11] with a **0.18** improvement in coherence score and reduces latency by **0.17s**, delivering smoother conversations with shorter wait times.

Due to system scheduling, all models maintain nearly the same response time of about 0.9s. Tab. 5 presents the detailed scores across six question types. Using hierarchical memory storage, our method excels in object search (OS), long-term memory search (LM), short-term memory search (SM), and conversational interaction (CI) tasks. Notably, our Slow model increases accuracy by **10.3%** in OS, **5.1%** in LM, **4.9%** in SM, and **5.8%** in CI compared to Video-online.

Table 4: **Quantitative results in StreamBench**. RPD is measured for streaming methods. Fr.: sampled frames.

| Method | FPS | Fr. | Sco. | Acc. | Coh. | RPD |
|--------|-----|-----|------|------|------|-----|
| Human performance | - | - | 4.03 | 79.4 | 1.16 | - |
| GPT-4o [27] | - | 50 | 3.70 | 71.0 | 1.66 | - |
| GPT-4o [27] | - | 35 | 3.64 | 69.8 | 1.72 | - |
| GPT-4o-mini [27] | - | 35 | 3.17 | 59.1 | 2.01 | - |
| *Instruct-tuning* | | | | | | |
| Video-LLaVA [4] | - | 8 | 2.81 | 48.9 | 2.19 | - |
| LLaMA-VID [2] | - | 180 | 2.94 | 51.2 | 2.08 | - |
| LLaVA-NExT [28] | - | 8 | 2.65 | 46.2 | 2.18 | - |
| LLaVA-Hound [29] | - | 8 | 3.12 | 54.7 | 1.83 | - |
| LongVA [20] | - | 8 | 3.05 | 52.4 | 1.96 | - |
| MiniCMP-v2.6 [30] | - | 8 | 2.97 | 56.6 | 2.21 | - |
| VILA1.5 [31] | - | 8 | 3.10 | 57.1 | 2.20 | - |
| InternVL2 [32] | - | 8 | 3.15 | 57.6 | 2.11 | - |
| InternLM-XCP2.5 [33] | - | 8 | 3.21 | 57.7 | 2.12 | - |
| *Training-free* | | | | | | |
| MovieChat [7] | - | 32 | 2.07 | 35.3 | 2.36 | - |
| FreeVA [8] | - | 4 | 3.10 | 56.3 | 2.11 | - |
| *Streaming* | | | | | | |
| Video-online [11] | 5 | - | 3.11 | 56.4 | 1.94 | 1.07 |
| Flash-VStream [10] | 1 | - | 2.89 | 52.1 | 2.21 | 4.15 |
| STREAMCHAT | | | | | | |
| Slow | 15 | - | **3.48** | **64.7** | **1.76** | 0.90 |
| Base | 20 | - | 3.42 | 63.8 | 1.79 | 0.89 |
| Fast | **32** | - | 3.28 | 61.7 | 1.81 | **0.85** |

**Offline Scenarios.** We compare our Base model against other methods in general offline video understanding benchmarks including MSRVTT-QA [12], ActivityNet [14], NExT-QA [15], MSVD-QA [16]. Since these benchmarks involve open-ended questions, we evaluate performances using score and accuracy as metrics, employing the same score model [3] as used in online tests. It should be noted that considering the limited average video length in the MSRVTT [12] and MSVD [16] shown in Table 1, we did not apply long-term memory $M_l$ for our model during the test. Additionally, since these open-ended question-answering test format benchmarks do not evaluate multi-round dialogue capabilities, we removed the dialogue memory $M_d$ component from our model.

Table 5: **Quantitative comparison across six tasks.** Detailed results for tasks 'OS', 'LM', 'SM', 'CI', 'KG', and 'SF'. For full names and definitions, refer to §2.1

| Method | Publication | OS | | LM | | SM | | CI | | KG | | SF | |
|---|---|---|---|---|---|---|---|---|---|---|---|---|---|
| | | Sco. | Acc. | Sco. | Acc. | Sco. | Acc. | Sco. | Acc. | Sco. | Acc. | Sco. | Acc. |
| Human performance | - - | 3.95 | 71.8 | 3.81 | 69.3 | 4.07 | 81.5 | 4.14 | 82.6 | 4.06 | 80.7 | 4.30 | 80.7 |
| GPT-4o-50 [27] | Arxiv 2024 | 3.27 | 60.5 | 3.35 | 61.2 | 3.41 | 64.4 | 3.81 | 72.3 | 4.58 | 93.9 | 3.83 | 74.7 |
| GPT-4o-35 [27] | Arxiv 2024 | 3.22 | 59.6 | 3.28 | 58.6 | 3.45 | 65.3 | 3.76 | 71.7 | 4.54 | 93.3 | 3.50 | 66.1 |
| GPT-4o-mini-35 [27] | Arxiv 2024 | 2.52 | 46.8 | 2.70 | 45.8 | 2.80 | 51.0 | 3.50 | 64.0 | 4.67 | 95.2 | 2.90 | 53.3 |
| *Instruct-tuning* | | | | | | | | | | | | | |
| Video-LLaVA [4] | EMNLP 2024 | 2.25 | 31.2 | 2.31 | 35.9 | 2.50 | 41.8 | 3.18 | 56.1 | 3.81 | 74.6 | 2.93 | 54.8 |
| LLaMA-VID [2] | ECCV 2024 | 2.32 | 33.9 | 2.43 | 38.2 | 2.63 | 44.1 | 3.31 | 58.4 | 3.93 | 76.9 | 3.06 | 57.1 |
| VILA1.5 [31] | CVPR 2024 | 2.33 | 36.1 | 2.54 | 44.3 | 2.87 | 50.8 | 3.59 | 68.3 | 3.97 | 78.6 | 3.38 | 65.5 |
| InternVL2 [32] | CVPR 2024 | 2.49 | 38.5 | 2.70 | 46.6 | 2.89 | 50.9 | 3.61 | 67.6 | 4.02 | 81.0 | 3.29 | 62.2 |
| LLaVA-NExT [28] | Arxiv 2024 | 2.17 | 35.0 | 2.14 | 31.4 | 2.15 | 36.0 | 2.55 | 42.7 | 3.88 | 76.1 | 3.12 | 57.6 |
| LLaVA-Hound [29] | Arxiv 2024 | 2.49 | 37.6 | 2.68 | 43.2 | 3.09 | 53.4 | 3.21 | 55.7 | 3.89 | 76.3 | 3.35 | 62.0 |
| LongVA [20] | Arxiv 2024 | 2.61 | 41.8 | 2.81 | 47.4 | 3.20 | 57.6 | 3.29 | 59.8 | 4.01 | 80.7 | 3.48 | 66.1 |
| MiniCMP-v2.6 [30] | Arxiv 2024 | 2.32 | 37.6 | 2.78 | 51.9 | 2.62 | 43.7 | 3.35 | 65.7 | 3.19 | 66.2 | 3.27 | 64.2 |
| InternLM-XCP2.5 [33] | Arxiv 2024 | 2.40 | 38.8 | 2.81 | 43.3 | 2.89 | 50.8 | 3.62 | 65.6 | 4.41 | 88.4 | 3.23 | 60.5 |
| *Training-Free* | | | | | | | | | | | | | |
| MovieChat [7] | CVPR 2024 | 1.45 | 18.6 | 1.42 | 20.4 | 1.76 | 26.5 | 2.28 | 42.3 | 3.39 | 67.2 | 2.05 | 35.8 |
| FreeVA [8] | Arxiv 2024 | 2.39 | 35.6 | 2.33 | 37.5 | 2.62 | 43.7 | 3.16 | 58.8 | 4.24 | 84.0 | 2.87 | 53.7 |
| *Online* | | | | | | | | | | | | | |
| Video-online [11] | CVPR 2024 | 2.61 | 41.4 | 2.87 | 48.8 | 3.01 | 52.9 | 3.31 | 62.7 | 3.58 | 69.2 | 3.39 | 64.1 |
| Flash-VStream [10] | Arxiv 2024 | 2.38 | 37.1 | 2.64 | 44.5 | 2.78 | 48.6 | 3.13 | 58.1 | 3.34 | 66.4 | 3.17 | 59.2 |
| STREAMCHAT | | | | | | | | | | | | | |
| Slow | - - | **3.01** | **51.7** | **2.93** | **53.9** | **3.21** | **57.8** | **3.86** | **68.5** | **4.38** | **88.1** | **3.57** | **69.3** |
| Base | - - | 2.93 | 50.5 | 2.87 | 52.9 | 3.15 | 56.1 | 3.82 | 67.6 | 4.37 | 87.9 | 3.56 | 68.8 |
| Fast | - - | 2.78 | 48.1 | 2.73 | 49.5 | 3.02 | 53.5 | 3.69 | 65.2 | 4.12 | 86.7 | 3.46 | 67.6 |

Table 6: **Performance comparison** of various models in offline video understanding benchmark.

| Method | Publication | ActNet | | NExT-QA | | MSVD | | MSRVTT | | Average | |
|---|---|---|---|---|---|---|---|---|---|---|---|
| | | Sco. | Acc. | Sco. | Acc. | Sco. | Acc. | Sco. | Acc. | Sco. | Acc. |
| Video-LLaVA [4] | EMNLP 2024 | 1.96 | 35.8 | 2.02 | 34.9 | 2.94 | 57.5 | 2.24 | 42.8 | 2.29 | 42.7 |
| LLaMA-VID [2] | ECCV 2024 | 2.09 | 36.6 | 2.07 | 36.0 | 2.83 | 56.9 | 2.23 | 42.6 | 2.30 | 43.1 |
| MovieChat [7] | CVPR 2024 | 2.27 | 37.8 | 2.05 | 35.6 | 2.97 | 57.9 | 2.15 | 43.0 | 2.36 | 43.5 |
| Video-online [11] | CVPR 2024 | 2.01 | 36.5 | 2.03 | 35.8 | 2.87 | 54.2 | 2.06 | 38.2 | 2.24 | 41.1 |
| LongVA [20] | Arxiv 2024 | 2.48 | 47.1 | 2.74 | 45.4 | 2.98 | 57.8 | 2.22 | 42.4 | 2.60 | 48.1 |
| LLaVA-Hound [29] | Arxiv 2024 | 2.69 | 48.7 | 2.56 | 43.7 | 3.07 | 56.8 | **2.42** | 42.7 | 2.68 | 47.9 |
| FreeVA [8] | Arxiv 2024 | 2.48 | 46.7 | 2.32 | 41.7 | 3.02 | 58.1 | 2.16 | 38.3 | 2.49 | 46.2 |
| Flash-VStream [10] | Arxiv 2024 | 2.02 | 37.3 | 2.06 | 36.1 | 2.91 | 56.1 | 2.08 | 39.8 | 2.26 | 42.3 |
| STREAMCHAT | - - | **2.78** | **50.1** | **2.84** | **50.5** | **3.08** | **58.7** | 2.38 | **43.4** | **2.77** | **50.6** |

- In the MSVD and MSRVTT benchmarks, our short-term memory module allows the model to capture more specific visual details, leading to accuracies of 58.7% and 43.4%, respectively.
- With the integration of long-term memory module, our model enhances the longer video performance, surpassing the previous best streaming method Flash-VStream [10] for **12.8%** and the best offline method LLaVA-Hound [29] for **1.4%** in ActivityNet [14] benchmark. In NExT-QA [15], our method can further improve the foundation model LongVA [20] by **5.1%** in accuracy.
- Although the base model LongVA [20] has achieved the best average accuracy in offline benchmarks, our method further improves it by **2.5%**, proving the effectiveness of the memory module.

## 4.3 CASE STUDY

In Fig. 6, we illustrate the reasoning process of STREAMCHAT with $g = 2$ to simplify the observation of internal mechanisms. The scenario involves a user asking STREAMCHAT to identify a tool meeting specific requirements and to describe its environment. The memory structure consists of a dialogue memory, $M_d$, with two historical entries, and a layered memory, $M_l$, with two levels. The memory tree visualization shows that the system initially searches for key information at level 1. It computes cosine similarity between the user's query $Q_i$ and two memory units, Summary (1) and (2), obtaining scores of 0.3993 and 0.4751, respectively. Based on these results, STREAMCHAT selects the path from the second node ($v_1$) due to its higher similarity score and continues along this path. Subsequently, the system aggregates value $\{v_r\}_{r=0}^{1}$ from $M_s$ into retrieved tokens that are then incorporated into the reasoning process. Additionally, a high similarity score of 0.6983 between $Q_i$ and the first historical conversation helps provide context, enhancing the depth and relevance of the response.

## 4.4 ABLATION STUDY

**Exploring Effects of Hierarchical Memory.** We conduct ablation experiments using the Base model to assess the impact of different memory components on performance. As shown in Table 7, adding $M_d$ to the base model improved performance on the CI task by 4.1% without affecting other

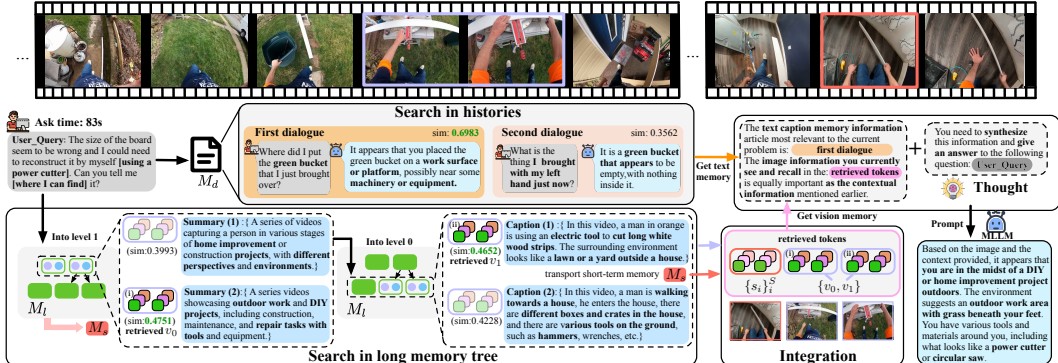

Figure 6: **An inference example of StreamChat** (§4.3). Given a question, our system retrieves the most related information in a long memory tree and dialogue histories based on the highest cosine similarity.

Table 7: **Analysis of hierarchical memory.** This table shows the impact of various memory configurations.

| $M_l$ | $M_s$ | $M_d$ | OS Sco. | OS Acc. | LM Sco. | LM Acc. | SM Sco. | SM Acc. | CI Sco. | CI Acc. | KG Sco. | KG Acc. | SS Sco. | SS Acc. | Average Sco. | Average Acc. |
|---|---|---|---|---|---|---|---|---|---|---|---|---|---|---|---|---|
| ✗ | ✗ | ✗ | 2.54 | 41.6 | 2.55 | 45.5 | 2.93 | 52.5 | 3.30 | 60.1 | **4.44** | **89.9** | **3.79** | **72.6** | 3.27 | 60.3 |
| ✗ | ✗ | ✓ | 2.55 | 41.9 | 2.55 | 45.7 | 2.94 | 52.5 | 3.66 | 64.2 | **4.44** | 88.7 | 3.78 | 72.4 | 3.32 | 60.9 |
| ✗ | ✓ | ✗ | 2.58 | 43.3 | 2.62 | 46.6 | 3.09 | 55.7 | 3.31 | 60.7 | 4.39 | 88.1 | 3.68 | 69.8 | 3.28 | 60.7 |
| ✓ | ✗ | ✗ | 2.85 | 49.5 | 2.78 | 51.7 | 2.96 | 53.5 | 3.32 | 61.1 | 4.42 | 88.4 | 3.65 | 69.4 | 3.33 | 62.2 |
| ✓ | ✓ | ✗ | 2.91 | 50.4 | **2.88** | **53.0** | 3.10 | 56.0 | 3.55 | 63.4 | 4.36 | 87.6 | 3.58 | 68.7 | 3.39 | 63.1 |
| ✓ | ✓ | ✓ | **2.93** | **50.5** | 2.87 | 52.9 | **3.15** | **56.1** | **3.82** | **67.6** | 4.37 | 87.9 | 3.56 | 68.8 | **3.42** | **63.8** |

tasks. Adding $M_l$ improved the LM task performance by 6.2%, while the use of $M_s$ boosts SM task performance by 3.2%. The results indicate that the model's performance in each subtask aligns with the inclusion of specific memory attributes. Additionally, we observe that different memory components can complement each other. When both long-term $M_l$ and short-term memory $M_s$ are applied simultaneously, the average accuracy increases by 0.9%.

**Tradeoffs in Speed and Threshold Settings.** The threshold of the Lucas-Kanade Optical Flow algorithm significantly influences video processing speeds. As illustrated in Fig. 7 (a), increasing the threshold initially accelerates the processing speed. However, this increase saturates when $t$ reaches 0.55, stabilizing at 32 FPS. Importantly, higher processing speeds are discouraged due to their detrimental impact on model performance (64.0%→60.7%). Elevating thresholds leads to more pronounced changes in frame differences and loss of original data, thereby limiting the model's ability to effectively utilize the full spectrum of video information.

**Design of Long Memory Tree.** The chunk length ($L$), group size ($g$), and clustering goal ($C$) significantly impact the effectiveness of the memory tree ($M_l$). In Fig. 7 (b-d), we evaluate how these factors influence online video understanding tasks, using the `Base` model with $t$=0.35.

- As shown in Fig. 7 (b), increasing $L$ form 15 to 30 leads to better performance (61.2%→64.0%). However, further increasing $L$ to 40 results in a slight decline (64.0%→63.1%), and substantially increases latency (0.84s→1.26s), due to the computational demands of the clustering algorithm.

- Increasing $g$ from 2 to 12, which represents the less compression of visual information and increases input sequence length $C \times L$, enhances performance (62.0%→63.9%) as greater diversity in the knowledge at each node of the long memory tree $M_l$ is achieved. However, it intensifies the load on the retrieval, leading to an increase in RDP (0.76s→1.02s), illustrated in Fig. 7 (c).

- The clustering goal ($C$) primarily influences the number of tokens ($v_i$) stored in the $M_l$. Fig. 7 (d) shows that increasing the dimension of $v_i$ (3→10) enhances model performance (59.4%→64.0%) by enriching the stored knowledge, which also exacerbates VRAM limitations (20→56 GB).

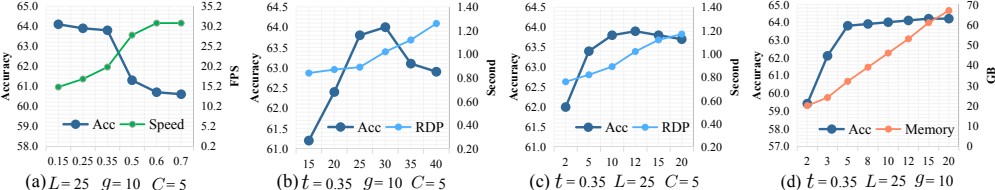

(a) $L=25$  $g=10$  $C=5$          (b) $t=0.35$  $g=10$  $C=5$          (c) $t=0.35$  $L=25$  $C=5$          (d) $t=0.35$  $L=25$  $g=10$

Figure 7: **Analysis of memory parameters**. (a) The influence between speed and threshold; Impact of (b) chunk length and (c) group size on performance and latency; (d) Effect of clustering goal on performance and VRAM.

## 5 RELATED WORK

**Multi-modal Language Models (MLMs).** Recent developments of large language models [3, 34–38] and multi-modal alignment techniques have significantly advanced MLMs' capability [39–44]. The LLaVA series [45, 46] utilizes straightforward mapping layers and visual instruction tuning to broaden image understanding tasks to video data. Challenges in video processing primarily involve efficiently compressing video within limited contextual windows. Innovations like ChatUniVi's [5] use of a K-NN clustering algorithm dynamically compress visual tokens, while LLaMA-VID [2] reduces single images to two tokens via cross-attention, and MovieChat [7] leverages long and short-term memory frameworks for extensive data handling. Despite these advances, the transition to effective real-time streaming video understanding in practical applications remains insufficiently addressed. Our research introduces a robust solution designed to meet the real-time demands of online video understanding, aiming to fill this critical gap.

**Streaming Video Understanding.** Streaming video understanding demands real-time responses from models to user queries, even as video durations potentially extend indefinitely. This is particularly challenging for traditional benchmarks like action recognition [47], multi-round video dialogue [48], and first-person question answering [13] which rely on uniform frame sampling. In response to these limitations, there is a growing shift towards online models that process only current and past video frames to formulate responses [10, 11]. Despite these advancements, these models often struggle with slow processing speeds and inadequate generalization capabilities, underlining a critical need for further exploration and enhancement in this field.

**Retrieval-Augmented Generation (RAG).** RAG combines information retrieval and text generation to produce more precise and informative responses by incorporating external knowledge into language models [49–59]. This technique has become increasingly popular for addressing knowledge retention and real-time information access challenges. MemoryBank [60] enhances interaction by storing real-time conversations and leveraging similarity search to retrieve contextually relevant information, enriching the depth and coherence of dialogue. This approach significantly improves a model's ability to maintain continuity in conversations, particularly in long or multi-turn interactions where maintaining context is crucial. Inspired by RAG's efficiency, we introduce a multi-modal memory system that integrates and updates textual and visual data in real time. Using a RAG-inspired retrieval mechanism, this system efficiently accesses the most relevant information from our memory bank, enabling the multi-modal language model to deliver precise, query-specific responses for enhanced video language understanding.

## 6 CONCLUSION

In this work, we introduce STREAMBENCH, a comprehensive benchmark specifically crafted to assess streaming video understanding, covering a broader range of video lengths and types with six question formats to simulate real-world human-robot interactions. This broader scope enhances our ability to evaluate model performance in complex and dynamic scenarios. Alongside, we present STREAMCHAT, a training-free method designed for efficient streaming video understanding, which treats video frames as compressible and storable units and manages them through a hierarchical memory structure. With advanced system scheduling, STREAMCHAT achieves real-time processing speeds and reduced interaction latency, demonstrating robust performance across both online and offline settings in our extensive experiments.

**Limitations and Future Works.** Our current retrieval algorithm relies on basic matching techniques, occasionally leading to incorrect responses. Enhancing this with more fine-grained retrieval mechanisms is an essential next step. Additionally, the VRAM constraints of our tree-structured storage could limit scalability as video duration and complexity further grow. Investigating more efficient or adaptive compression techniques will address these limitations. Moreover, to achieve lower latency, we plan to explore closer hardware integration and the potential adoption of fast-serving, multi-modal distributed systems to accommodate larger model parameters and increased user demands.

## ACKNOWLEDGEMENTS

This work was supported in part by the National Natural Science Foundation of China under Grants 62406053, 62441231 and 62206039, and in part by the Fundamental Research Funds for the Central Universities under Grants DUT24YG119 and DUT24RC(3)025.

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

## SUMMARY OF THE APPENDIX

This appendix contains additional details for the ICLR 2025 submission, titled '*Streaming Video Understanding and Multi-round Interaction with Memory-enhanced Knowledge*', which is organized as follows:

- §A presents the dataset collection pipeline.
- §B visualizes more cases of our benchmark.
- §C offers the details about retrieval algorithm.
- §D introduces the details and calculation method of metrics.
- §E shows failure cases and analysis.
- §F discusses model selection and deployment strategies.
- §G outlines our plans for benchmark expansion.

## A    DATA PIPELINE

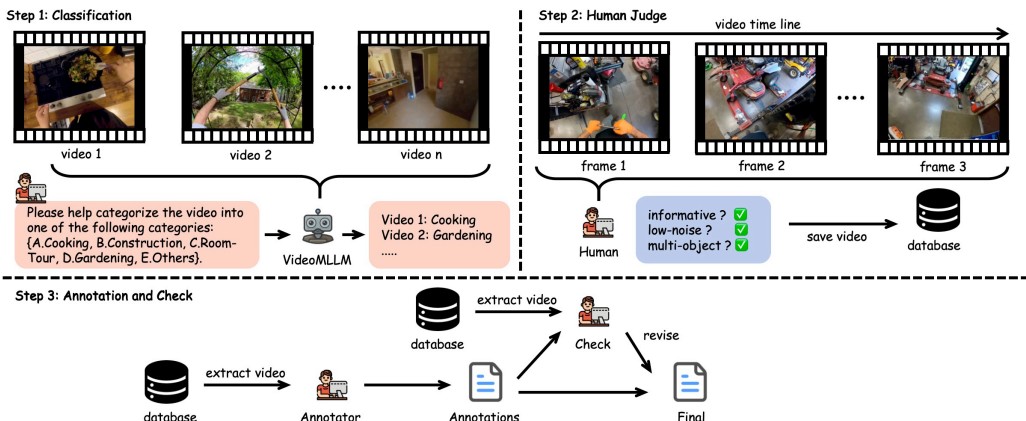

Figure 8: The date preparation pipeline utilized in STREAMBENCH.

Fig. 8 presents our video collection pipeline. It consists of 3 parts: (1) Classification; (2) Human judge; and (3) Annotation check. First, a MLLM [4] is utilized to complete the video classification based on our requirements. The following prompt is used during the first data filtering step:

``` 
" Based on the observed video information, categorize the video
into one of the predefined categories listed in {All_Classes}.
Respond exclusively in the format of a Python dictionary string
with the keys 'pred' and 'score'.  The 'pred' key should contain
the uppercase STRING of the chosen category.  Refrain from
providing any additional text or explanatory output.  Your
response should strictly follow this example:  {'pred': 'A'}."
```

{All_Class} is the options formation. When dealing with different datasets, we need to change the options. For example, when dealing with Youtube-8M [19], it is {A: Drama, B: Action, C: Cartoon, D: Romance, E: Sci-fi, F: Others} and {A: Cooking, B: Construction, C: Room-Tour, D: Gardening, E: Others} for EgoSchema [13] dataset. We save the output to a JSON file and then find categories from the file as needed. It is worth noting that we also used the original category information in the YouTube data. The above classification process is used primarily for secondary classification of MovieClips data. For EgoSchema, we need to classify all original videos as they don't contain category annotations.

## B    MORE VISUALIZATIONS

In Fig. 9-12, we visualize several STREAMCHAT cases applied to different types of videos. Specifically, Fig. 9 illustrates an egocentric video annotation, where the system engages in interactive questioning

based on the visual information captured from a first-person perspective. In this figure, various types of annotation questions are showcased, including object identification, memory recall, and knowledge-based interactions. Each example contains six distinct questions, with the Simple Start (SS) question placed first, while the sequence of the remaining five questions varies randomly throughout the video. The questions explore aspects such as short-term memory, long-term memory, object search, and conversational interaction, allowing for a broad range of analysis. To further enhance clarity, specific frames are highlighted along with the locations of key objects referenced in the questions, enabling a better understanding of how the system interacts with the visual context at different moments.

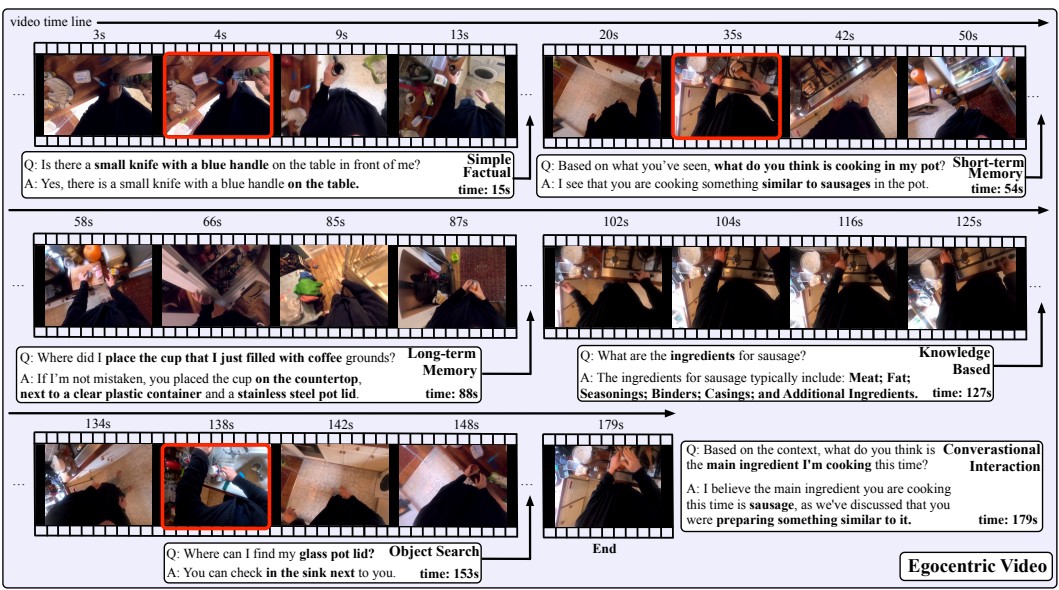

Figure 9: Visualization of egocentric video analysis.

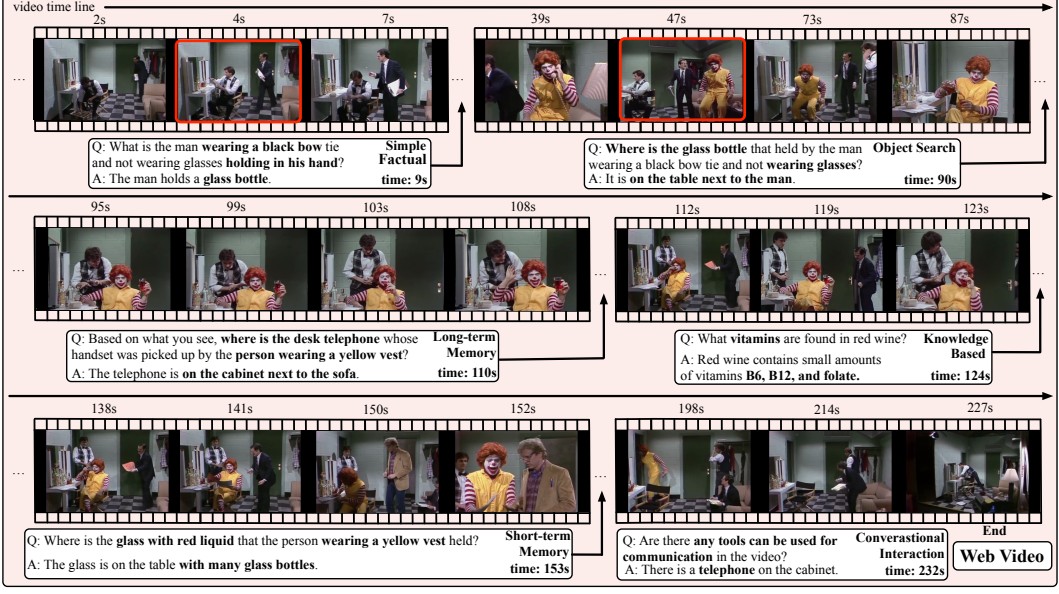

Figure 10: Visualization of web video analysis.

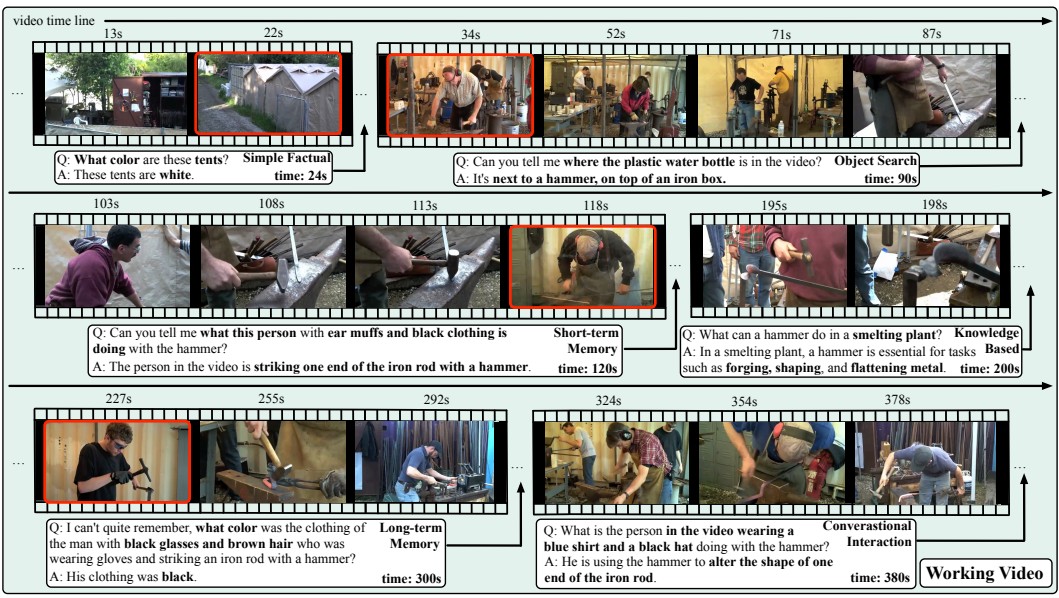

Figure 11: Visualization of working video analysis.

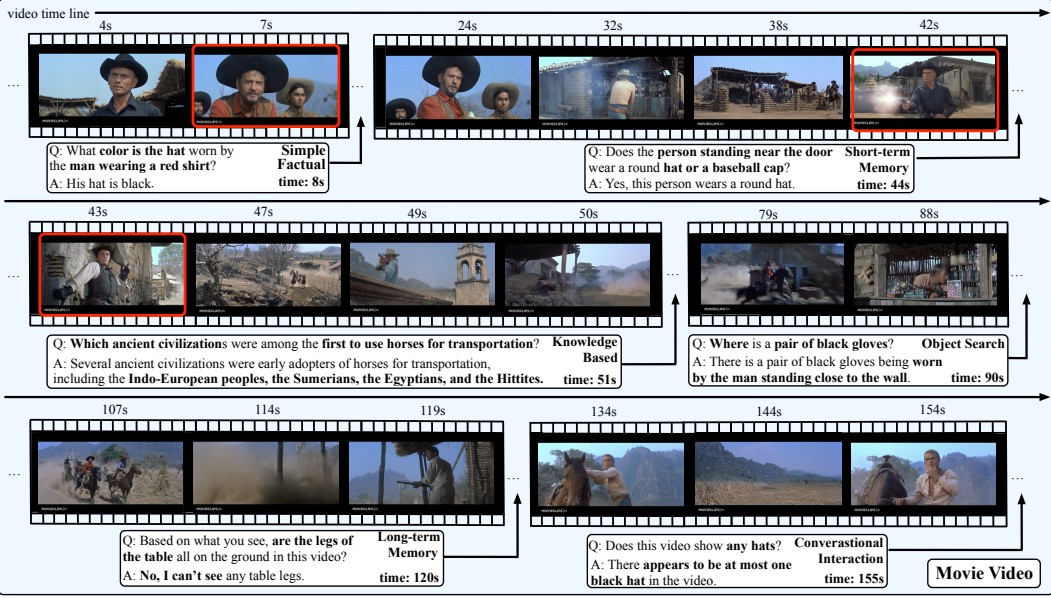

Figure 12: Visualization of movie video analysis.

## C RETRIEVAL ALGORITHM

Inspired by the retrieval argumentation system [49], our approach enhances the model's capability to address complex queries by retrieving the most relevant information from long-term memory for contextual support. As outlined in Algorithm 1, we compute the similarity, Sim, between the user's request, $Q$, and entries $T_n$ in the memory. This process identifies the optimal path for accessing the most pertinent stored knowledge, $T_{best}$ and $C_{best}$. Leveraging our tree-like storage structure, we efficiently focus on the highest-similarity nodes at each layer, minimizing the computational load by avoiding exhaustive sub-node calculations. The selected knowledge, $T_{best}$ and $C_{best}$, is then

integrated with $Q$ in a mixed prompt format to serve as the final input for the multi-modal language model, facilitating accurate response generation.

---

**Algorithm 1** Knowledge Retrieval from Long-Term Memory

---

1:  **Input:** User request $Q$
2:  **Output:** Best matching knowledge $T$ and $C$
3:  Initialize similarity $Sim_{max} \leftarrow -\infty$
4:  Initialize best path knowledge $T_{best}, C_{best} \leftarrow \emptyset, \emptyset$
5:  **for** each node $n$ in the tree structure **do**
6:      Compute similarity $Sim(Q, T_n)$ where $T_n$ is the caption at node $n$
7:      **if** $Sim(Q, T_n) > Sim_{max}$ **then**
8:          $Sim_{max} \leftarrow Sim(Q, T_n)$
9:          $T_{best}, C_{best} \leftarrow T_n, C_n$                    ▷ Update best match knowledge
10:     **end if**
11:     **if** node $n$ has children **then**
12:         Continue to next level
13:     **end if**
14: **end for**
15: Reconstruct input for MLLM using $T_{best}, C_{best}$, and $Q$
16: **return** $T_{best}, C_{best}$

---

## D  DETAILS OF METRICS

We design the following metrics to measure the model's ability to stream video understanding:

**(1) Score and Accuracy**: To assess the semantic correctness of a single-turn dialogue, using language models is a mainstream approach [12, 14, 16–18]. We also use this as a key metric in our benchmark. In our test benchmark, we use the open-source language model LLaMA-3 8B [3] Instruct version as our scoring model $f$. Here is the prompt that we used during scoring:

```
Prompt = [{"role":  "system", "content": "You are an intelligent chatbot designed
for evaluating the correctness of generative outputs for question-answer pairs.
Your task is to compare the predicted answer with the correct answer and determine if they match
meaningfully. Here's how you can accomplish the task:
INSTRUCTIONS:
- Focus on the meaningful match between the predicted answer and the correct answer.
- Consider synonyms or paraphrases as valid matches.
- Evaluate the correctness of the prediction compared to the answer. "}
{"role":  "system", "content": "Please evaluate the following video-based question-
answer pair: Question: question; Correct Answer: answer; Predicted Answer: prediction
Provide your evaluation only as a yes/no and score where the score is an integer value between 0 and
5, with 5 indicating the highest meaningful match.
Please generate the response in the form of a Python dictionary string with keys 'llama pred' and
'score', where the value of 'llama pred' is a string of 'yes' or 'no' and the value of 'score' is in
INTEGER, not STRING. DO NOT PROVIDE ANY OTHER OUTPUT TEXT OR EXPLANATION.
Only provide the Python dictionary string. For example, your response should look like this: {'llama
pred': YOUR JUDGE, 'score': YOUR SCORE.}" } ]
```

Table 8: Prompt given to the LLaMA-3 model for evaluation.

We organize the question $Q$, reference answer $R$, and model's response $M$ into the Tab. 8 formation and send to scoring model, which then provides a score in the range of 0-5 and evaluates whether the model's response is semantically correct:

$$S_i = f(Q, R, M), Acc = \frac{1}{N} \sum_{i=1}^{N} \mathbb{I}(S_i \geq T) \tag{6}$$

A higher score $S_i$ and $Acc$ indicates that the answer is closer to the reference answer.

**(2) Coherence**: Given that a single video may involve multiple rounds of dialogue, we need to evaluate the model's ability to provide a coherent experience across different rounds. We introduce

the coherence metric, which calculates the absolute value of the difference between the semantic scores $S_i$ of different dialogues within a single scenario. The average of all these differences is used as the coherence metric. The calculation formula is as follows:

$$C = \frac{1}{N-1} \sum_{i=1}^{N-1} |S_i - S_{i+1}|, \tag{7}$$

where $C$ is the coherence score, $N$ is the total number of dialogue turns in the scenario, $S_i$ represents the semantic score of the $i$-th dialogue turn and $|.|$ is the absolute difference between the semantic scores of consecutive dialogue turns. It is evident that a smaller $C$ indicates that the model provides a better coherence experience for the user.

**(3) Request Processing Delay**: For online scenarios, system latency consists of two parts: 1. Request processing delay; 2. Generation delay. The generation delay is mainly influenced by factors such as context length, language model parameters, and deployment methods, and can be adjusted through various methods. In this benchmark, we primarily assess (1) request processing delay, which is calculated as the time from when the user completes the request input to when the model starts generating the response. The calculation formula is as follows:

$$\text{RPD} = T_{\text{start}} - T_{\text{input}} \tag{8}$$

## E  FAILURE CASE AND ANALYZE

We present some failure cases that occurred during testing and explain why they occurred. Most of these cases come from object search, long-term memory, short-term memory and conversational interaction tasks. The problems that occurred are mainly grouped into four types:

- **Temporal Fine-grained:** In the object search task, our method still struggles to identify key information when the queried objects or events appear too briefly or sporadically. For instance, as demonstrated in case (1), the user's question pertains to a candle. However, due to the infrequent appearance of the candle and its small size, the model fails to provide an accurate response.
- **Spatial Fine-grained:** Whether in object search or short-term memory task, our method faces limitations when the user's target is too small or blends into the background, even if the object appears multiple times in the video. For example, in short-term memory task case (2) and object search task case (3), the target objects (porcelain bowl and red cup) are too small relative to the foreground, making it difficult for the model to accurately detect and locate them. We will continue to improve our method to enhance the perception of small objects.
- **Target Movement:** During the reasoning process, we observed that in some cases, even when the model correctly identified the target, its interpretation of the target's actions and relationships with surrounding objects was still inaccurate. For instance, in the long-term memory task case (4) the model failed to recognize the action and position association between the "person" and the "box," leading to an incorrect response.
- **Context Induction:** In the conversational interaction task, the model's performance is influenced by the accuracy of its responses to previous related questions. For instance, in case 5, the model retrieved information from the dialogue history, but when that historical information was incorrect, it became challenging for the model to provide the correct answer.
- **Information Loss:**  According to the experimental results in Tab. 5, although our method shows balanced performance in various tasks of StreamChat, our hierarchical memory storage still has the potential risk of losing information. Since our existing method is too dependent on the accuracy of the retrieval algorithm, we will continue to update our method to minimize information loss.

## F  MODEL SELECTION AND DEPLOYMENT

Our research indicates that a suitable model should possess the following key attributes:

- **Long-Form Video Understanding:** Effective processing of long videos is crucial. While we utilize K-Means for feature compression, the information retrieved by our memory mechanism remains relatively long, requiring a model capable of handling extended sequences.
- **Robustness to Prompt Variations:** For accurate and coherent multi-turn conversations, the model must be robust to changes in prompt wording. This robustness is essential to prevent inconsistencies

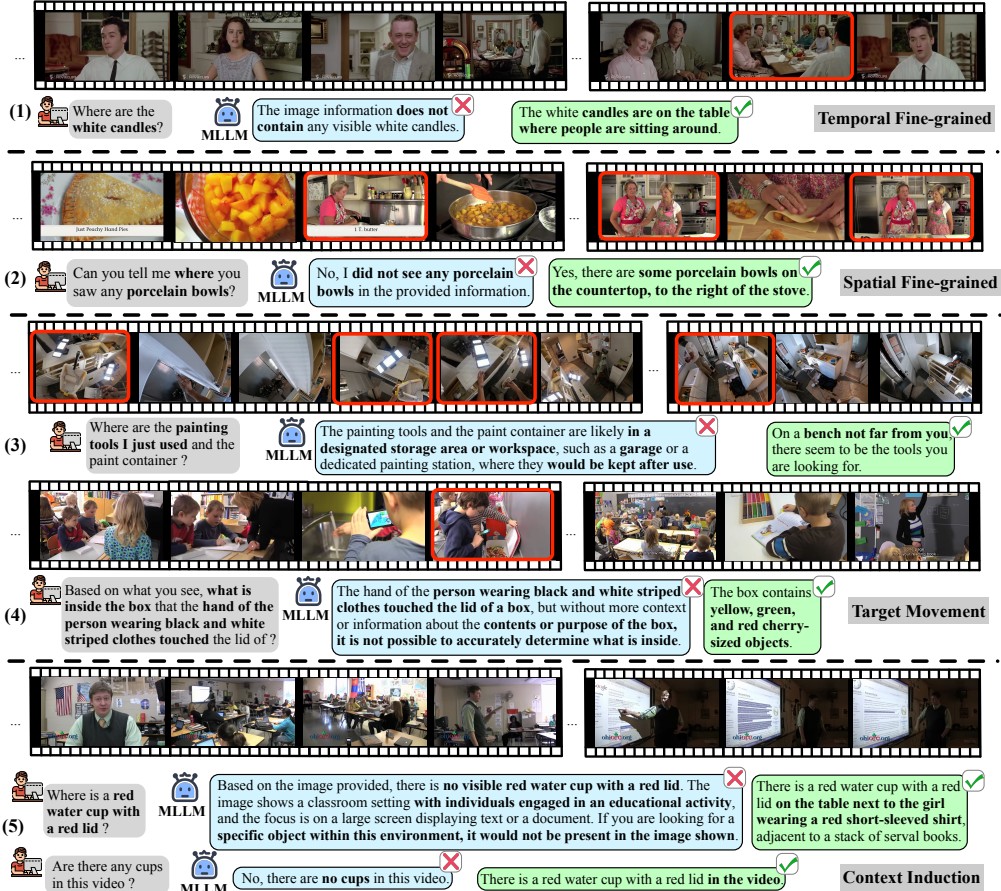

Figure 13: Visualization of failure cases.

or hallucinations in the model's output when prompts are adjusted to incorporate information from the memory mechanism.

By integrating LongVA with our proposed system, we successfully extend its capabilities to encompass streaming video processing and multi-turn conversations while preserving these critical characteristics. As we introduced in §4, we utilize 2 GPUS to complete the deployment of our method. The main reason is that during system scheduling, we need to utilize tensor parallelism to distribute the computational load for efficient execution. Specifically, the *(i) selective frame stacking thread* and *(iii) context summarization thread* are running on GUP1 while *(ii) memory formation thread* is running on GPU2.Therefore, the compressed video tensors need to be transmitted between different GPUs to ensure the stable operation of the system.

# G  EXPANSION PLAN OF STREAMBENCH

- **Video scale:** We are trying to expand the number of videos contained in the StreamBench to reach a higher standard. We are working on expanding the number of videos to **thousands** while maintaining the diversity of video length and types.
- **Annotation scale:** We are continuing to promote the development of high-quality annotations. Based on your suggestions, we will use manual annotation methods to expand the annotation of existing benchmarks to the order of **ten thousand levels** and also use manual inspection to filter out toxic labels and erroneous information.
- **Diverse tasks:** Given that the current benchmark only has a single task type, we are continuing to expand the types of tasks included in the benchmark, including but not limited to **multiple-choice questions, video captioning, and video grounding and etc.**

