# OpenReview forum: "Streaming Video Understanding and Multi-round Interaction with Memory-enhanced Knowledge"
_ICLR.cc/2025/Conference — ICLR 2025 Poster_

### Official Review · Reviewer_yVyo · 2024-11-01

**Soundness:** 2
**Presentation:** 2
**Contribution:** 2
**Rating:** 3
**Confidence:** 4

**Summary:**

This paper studies streaming video understanding. The authors propose a new benchmark named StreamBench to evaluate streaming video understanding across diverse media types and interactive scenarios, including multi-turn interactions and complex reasoning tasks. They also propose StreamChat, a training-free framework for streaming video reasoning and conversational interaction, with a complex memory mechanism. Their method enhances processing speed and reduces latency, ensuring robust performance in real-world applications. Extensive evaluations on StreamBench and some public benchmarks demonstrate that StreamChat outperforms the selected
baselines.

**Strengths:**

1. The proposed StreamBench may be the first benchmark for streaming video understanding.
2. The proposed StreamChat outperforms the selected baselines on StreamBench.
3. The processing speed of StreamChat significantly outperforms those of baselines.

**Weaknesses:**

1. The proposed dataset is too small, only 306 videos and 1.8K  question-answer pairs are collected. The current video benchmark typically has at least thousands of videos and tens of thousands of QA pairs. This version of the benchmark is not ready for release.
2. State-of-the-art video LLMs are not included in the benchmark, such as MiniCPM-V 2.6 [1], InternLM-XComposer2.5 [2], VILA [3], and InternVL2 [4]. The effectiveness of the proposed method is unclear.
3. The Selective Frame Stacking seems may ignore small objects in the video and only focus on the global frame feature.
4. The proposed memory mechanism is very complicated and discards a large amount of information in the video.
5. The authors test the processing speed of models on two NVIDIA Tesla A800 80GB GPUs, which is not a typical scenario of model development in real-world applications.


[1] Yao, Y., Yu, T., Zhang, A., Wang, C., Cui, J., Zhu, H., ... & Sun, M. (2024). Minicpm-v: A gpt-4v level mllm on your phone. arXiv preprint arXiv:2408.01800.
[2] Zhang, P., Dong, X., Zang, Y., Cao, Y., Qian, R., Chen, L., ... & Wang, J. (2024). Internlm-xcomposer-2.5: A versatile large vision language model supporting long-contextual input and output. arXiv preprint arXiv:2407.03320.
[3] Lin, J., Yin, H., Ping, W., Molchanov, P., Shoeybi, M., & Han, S. (2024). Vila: On pre-training for visual language models. In Proceedings of the IEEE/CVF Conference on Computer Vision and Pattern Recognition (pp. 26689-26699).
[4] Chen, Z., Wang, W., Tian, H., Ye, S., Gao, Z., Cui, E., ... & Qiao, Y. (2024). How far are we to gpt-4v? closing the gap to commercial multimodal models with open-source suites. arXiv preprint arXiv:2404.16821.

**Questions:**

Please respond to Weaknesses.

---

> ### Author Response · Authors · 2024-11-23
> **Official Comment by Authors**
>
> **Thank you for your insightful feedback. We appreciate your  your recognition of the inference efficiency of StreamChat and the novelty of StreamBench. Our detailed responses to your concerns are provided below:**
>
> **Q1: The proposed dataset is too small, only 306 videos and 1.8K question-answer pairs are collected.**
>
> **A1:** We fully understand your concerns about the size of the dataset, and we also considered many factors when creating benchmark data.
>
> **First, because we are focusing on streaming video scenarios, we pay more attention to the average video length.** As shown in the analysis results in Section 2, our video length currently reaches 276s, which exceeds most video benchmarks. Longer time brings more lengthy video information and also increases the difficulty of the task. Even so, we maintain the average length while also ensuring the total length of the video. For example, compared with MovieChat benchmark[9], we have more videos.
>
> **Second, we spent a lot of effort to complete the annotation.** All the questions and answers currently used are from manual annotation and are also manually cleaned. It took us a total of 42 days to complete the annotations. Although there are only 1.8k questions, these questions are carefully designed. Compared with the previous benchmark that used machine-assisted annotation like Video-online[1] and Flash-VStream[2], we believe that this approach can ensure sufficient quality.
>
> **Third, we ensure the diversity of the benchmark.** As analyzed in figure 2 in Section 2, StreamBench shows diversity in video categories, number of questions, and video duration distribution, thus ensuring the fairness of the benchmark.
>
> **We appreciate your suggestions and we will continue to develop the StreamBench in the following aspects:**
>
> **1. Video scale.** We will further expand the number of videos contained in the StreamBench to reach a higher standard. **We will take your suggestions and expand the number of videos to thousands while maintaining the diversity of video length and types.**
>
> **2. Annotation scale.** We will continue to promote the development of high-quality annotations. **Based on your suggestions, we will use manual annotation methods to expand the annotation of existing benchmarks to the order of ten thousand levels and also use manual inspection to filter out toxic labels and erroneous information.**
>
> **3. Diverse tasks.** Given that the current benchmark only has a single task type, we will continue to expand the types of tasks included in the benchmark, including but not limited to **multiple-choice questions, video captioning, and video grounding and etc.**
>
> At the same time, **we also provide a clear data expansion plan in the revised version of the manuscript (see Section 2(L193-L194) and Appendix G (L1020-L1030)).**

---

> ### Author Response · Authors · 2024-11-23
> **Official Comment by Authors**
>
> **Q2: Performance of state-of-the-art Video LLMs in StreamBench.**
>
> **A2:** Thank you for your suggestion, measuring these more powerful models will make our paper more complete. To address your concerns, we have supplemented the results of  MiniCPM-V 2.6 [3], InternLM-XComposer2.5 [4], VILA [5], and InternVL2 [6] on StreamBench and added them to the paper.
>
> | Method | FPS | Fr. | Sco. | Acc. | Coh. | RPD |
> | ---- | ---- | ---- | ---- | ---- | ---- | ---- |
> | Video-LLaVA        | -- |  8 | 2.81 | 48.9 | 2.19 |  --  |
> | LLaMA-VID          | -- | 180| 2.94 | 51.2 | 2.08 |  --  |
> | LLAVANExT          | -- |  8 | 2.65 | 46.2 | 2.18 |  --  |
> | LLAVA-Hound        | -- |  8 | 3.12 | 54.7 | 1.83 |  --  |
> | LongVA             | -- |  8 | 3.05 | 52.4 | 1.96 |  --  |
> | MovieChat          | -- | 32 | 2.07 | 35.3 | 2.36 |  --  |
> | FreeVA             | -- |  4 | 3.10 | 56.3 | 2.11 |  --  |
> | **MiniCMP_v2.6**   | -- |  **8** | **2.97** | **56.6** | **2.21** |  --  |
> | **VILA1.5**            | -- |  **8** | **3.10** | **57.1** | **2.20** |  --  |
> | **InternVL2**          | -- |  **8** | **3.15** | **57.6** | **2.11** |  --  |
> | **InternLM-XCP2.5** | -- | **8** | **3.21** | **57.7** | **2.12** |  --  |
> | Video-online       | 5  | -- | 3.11 | 56.4 | 1.94 | 1.07 |
> | Flash-VStream      | 1  | -- | 2.89 | 52.1 | 2.21 | 4.15 |
> | StreamChat-Slow    | 15 | -- | 3.48 | 64.7 | 1.76 | 0.90 |
> | STREAMCHAT-Base    | 20 | -- | 3.42 | 63.8 | 1.79 | 0.89 |
> | STREAMCHAT-Fast    | 32 | -- | 3.28 | 61.7 | 1.81 | 0.85 |
>
>
> | ~ | OS | LM | SM | CI | KG | SF |
> | --- | --- | --- | --- | --- | --- | --- |
> | Method  | Sco./Acc. | Sco./Acc. | Sco./Acc. | Sco./Acc. | Sco./Acc. | Sco./Acc. |
> | Video-LLaVA               | 2.25/31.2 | 2.31/35.9 | 2.50/41.8 | 3.18/56.1 | 3.81/74.6 | 2.93/54.8 |
> | LLaMA-VID                 | 2.32/33.9 | 2.43/38.2 | 2.63/44.1 | 3.31/58.4 | 3.93/76.9 | 3.06/57.1 |
> | LLaVA-NExT                | 2.17/35.0 | 2.14/31.4 | 2.15/36.0 | 2.55/42.7 | 3.88/76.1 | 3.12/57.6 |
> | LLaVA-Hound               | 2.49/37.6 | 2.68/43.2 | 3.09/53.4 | 3.21/55.7 | 3.89/76.3 | 3.35/62.0 |
> | LongVA                    | 2.61/41.8 | 2.81/47.4 | 3.20/57.6 | 3.29/59.8 | 4.01/80.7 | 3.48/66.1 |
> | **MiniCMP_v2.6**          | **2.32/37.6** | **2.78/51.9** | **2.62/43.7** | **3.35/65.7** | **3.19/66.2** | **3.27/64.2** |
> | **VILA1.5**               | **2.33/36.1** | **2.54/44.3** | **2.87/50.8** | **3.59/68.3** | **3.97/78.6** | **3.38/65.5** |
> | **InternVL2**             | **2.49/38.5** | **2.70/46.6** | **2.89/50.9** | **3.61/67.6** | **4.02/81.0** | **3.29/62.2** |
> | **InternLM-XCP2.5** | **2.40/38.8** | **2.81/43.3** | **2.89/50.8** | **3.62/65.6** | **4.41/88.4** | **3.23/60.5** |
> | MovieChat                 | 1.45/18.6 | 1.42/20.4 | 1.76/26.5 | 2.28/42.3 | 3.39/67.2 | 2.05/35.8 |
> | FreeVA                    | 2.39/35.6 | 2.33/37.5 | 2.62/43.7 | 2.94/54.7 | 4.24/84.0 | 2.87/53.7 |
> | Video-online   | 2.61/41.4 | 2.87/48.8 | 3.01/52.9 | 3.31/62.7 | 3.58/69.2 | 3.39/64.1 |
> | Flash-VStream  | 2.38/37.1 | 2.64/44.5 | 2.78/48.6 | 3.13/58.1 | 3.34/66.4 | 3.17/59.2 |
> | StreamChat-Slow| 3.01/51.7 | 2.93/53.9 | 3.21/57.8 | 3.86/68.5 | 4.38/88.1 | 3.57/69.3 |
> | StreamChat-Base| 2.93/50.5 | 2.87/52.9 | 3.15/56.1 | 3.82/67.6 | 4.37/87.9 | 3.56/68.8 |
> | StreamChat-Fast| 2.78/48.1 | 2.73/49.5 | 3.02/53.5 | 3.69/65.2 | 4.12/86.7 | 3.46/67.6 |
>
> These results provide a more comprehensive comparison and further demonstrate the effectiveness of StreamChat, particularly in terms of its improved performance and efficiency trade-off.  **We have updated the experiment results (see Section 4 Table 4 and Table 5) in the revised version accordingly.  Please let us know if you want to see more experimental results.**

---

> ### Author Response · Authors · 2024-11-23
> **Official Comment by Authors**
>
> **Q3: The Selective Frame Stacking module ignores small objects.**
>
> **A3:** We appreciate your question regarding the Selective Frame Stacking module and its potential impact on small object perception and understanding.  The Selective Frame Stacking is not designed to solely focus on global features, and we believe it remains sensitive to small objects.
>
> **First, the primary function of Selective Frame Stacking is to calculate motion vectors between consecutive frames and selectively encode frames based on a motion threshold.**  As illustrated in Figure 4, this primarily improves computational efficiency by reducing redundant processing of nearly identical frames, such as those depicting the same person performing similar actions.
>
> **Second, StreamChat achieves its best performance on the Object Search (OS) task within StreamBench.** This task, as described in Section 2.2, specifically requires locating objects that appear briefly (<5s) within the video.  This demonstrates StreamChat's ability to perceive and locate even small, fleeting objects.  Examples of such questions are provided in Appendix B (e.g., the second question in Figure 11).
>
> **Third, we provide an inference example of StreamChat as shown in Figure 6 to demonstrate its perception of small objects.** In this example, our goal is to find tools that appear in the video, and according to the inference process, our method is able to solve this problem.
>
> **According to your suggestion, we supplement the description of StreamChat’s small object perception capabilities in Appendix E (L939-L967) of the revised version.**
>
> **Q4: Information discard and complicity of memory mechanism.**
>
> **A4:** We appreciate your feedback regarding the complexity of our hierarchical memory mechanism and the potential information loss.
>
> The system scheduling module (Figure 5) mitigates complexity by decoupling memory components into separate threads, enabling low latency (<1s) and real-time processing (30FPS).  This allows the memory mechanism to **operate efficiently despite its multi-component design.**
> While the K-Means algorithm used in the long memory tree introduces potential information loss, ablation studies show that this is balanced by overall performance gains.  StreamChat maintains stable performance across various StreamBench tasks (LM, SM) and offline question-answering scenarios.
>
> **However, we acknowledge and appreciate your suggestion and enriched our paper about the shortcomings of our memory mechanism (see failure case and analyze in Appendix E (L939-L967) ) in the revised version.**
> Meanwhile, we are actively exploring improvements to our memory mechanism, including:
>
> **1. Information Compression:** Investigating more robust and dynamic methods like MADTP [10] and IVTP [11] as alternatives to K-Means.
>
> **2. Information Storage:** Expanding the stored information beyond compressed visual features and conversation history to include richer textual descriptions and auxiliary information.
>
> **3. Retrieval Methods:** Exploring more advanced retrieval algorithms like Raptor [12] and MemWalker [13] to reduce reliance on text encoding models and improve retrieval accuracy.

---

> > ### Author Response · Authors · 2024-11-23
> > **Official Comment by Authors**
> >
> > **Q5: The authors test the processing speed of models on two NVIDIA Tesla A800 80GB GPUs, which is not typical.**
> >
> > **A5:** We acknowledge that evaluating inference speed on two NVIDIA Tesla A800 80GB GPUs is not typical for real-world applications.  **Our principle for this setup stems from the large number of parameters in StreamChat and the need for efficient execution.**  Similar to large language model serving systems like vLLM [7] and SGLang [8], StreamChat utilizes tensor parallelism to distribute the computational load and improve throughput.  For instance, SGLang employs tensor parallelism to efficiently serve models like LLaVA-NExT-Video-34B. While we recognize that our current two-GPU implementation may not ideal for typical users, it allows us to explore the potential of StreamChat in real-time processing.
> >
> > **Based on your suggestions, we have explained the reasons and details of our deployment in the revised version (in Section 4(L343-344) and Appendix F (L1011-L1018) ) to facilitate more clearly understanding of our approach.**
> >
> > **1. System scheduling optimization.** Improving VRAM utilization, drawing inspiration from efficient serving technologies like SGLang and vLLM.
> >
> > **2. Model quantization.** Employing techniques like AWQ [14] and SmoothQuant [15] to reduce model size and enable single-GPU operation.
> >
> > **3. Efficient memory artecture.**  Simplifying our memory storage mechanism to further improve efficiency and broaden deployment scenarios.
> >
> > We hope these explanations address your concerns.  Please let us know if you have any further questions.
> >
> > [1] Videollm-online: Online video largelanguage model for streaming video. CVPR 2024
> >
> > [2] Flash-vstream: Memory-based real-time understanding for long video streams. arXiv 2024.06
> >
> > [3] MiniCPM-V: A GPT-4V Level MLLM on Your Phone. arXiv 2024.08
> >
> > [4] InternLM-XComposer-2.5: A Versatile Large Vision Language Model Supporting Long-Contextual Input and Output. arXiv 2024.07
> >
> > [5] VILA: On Pre-training for Visual Language Models. CVPR 2024
> >
> > [6] InternVL: Scaling up Vision Foundation Models and Aligning for Generic Visual-Linguistic Tasks. CVPR 2024
> >
> > [7] Efficient memory management for large language model serving with pagedattention. ACM SOSP 2023
> >
> > [8] SGLang: Efficient Execution of Structured Language Model Programs. github 2024
> >
> > [9] Moviechat: From dense token to sparse memory for long video understanding. CVPR 2024
> >
> > [10] MADTP: Multimodal Alignment-Guided Dynamic Token Pruning for Accelerating Vision-Language Transformer. CVPR 2024
> >
> > [11] IVTP: Instruction-guided Visual Token Pruning for Large Vision-Language Models. ECCV 2024
> >
> > [12] Raptor: Recursive abstractive processing for tree-organized retrieval. ICLR 2024
> >
> > [13] Walking down the memory maze: Beyond context limit through interactive reading. arXiv 2023.10
> >
> > [14] AWQ: Activation-aware Weight Quantization for On-Device LLM Compression and Acceleration. Mlsys 2024
> >
> > [15] SmoothQuant: Accurate and Efficient Post-Training Quantization for Large Language Models. Mlr 2023

---

> ### Comment · Reviewer_yVyo · 2024-11-25
> **Reply to Authors**
>
> Thanks for the authors' reply. However, I still have some unresolved concerns.
>
> For Q1, I understand the difficulty of constructing a streaming video QA dataset. The current dataset containing 306 videos and 1.8K question-answer pairs is too small. I look forward to your expanded dataset.
>
> For Q2, while the authors have added all baselines I mentioned, I agree with Reviewer 9hDR's concern about the evaluation method. Why is LLaMA-3 chosen for evaluation? Why not other LLMs, especially as there are LLMs stronger than LLaMA-3 (such as GPT-4o)? And are the evaluation results of LLaMA-3 aligned with human's judgment? The rationale behind the current evaluation method is not clear enough.
>
> For Q4, I agree with Reviewer 9hDR's concern. The hierarchical memory design seems a bit ad-hoc. The motivation behind the architecture design is not clear enough.
>
> For Q5, the authors' reply makes me more confused about this experimental setting. This setting seems to be specially optimized to deploy the authors' model, which makes the efficiency comparison with other models unfair.

---

> > ### Author Response · Authors · 2024-11-25
> > **Official Comment by Authors**
> >
> > Thanks for your prompt feedback. We address your concerns in the following statement:
> >
> > **Q1: Why use LLaMA-3 to evaluate predictions?**
> >
> > **A1:** There are mainly two reasons:
> >
> > **First, LLaMA-3 is not only one of the most powerful language models , but also has a complete open source attribute.** This is very conducive to more researchers to make further improvements and reproduce our results under cost-controlled conditions.
> >
> > **Second, according to our experimental results, we found that LLaMA-3 and human judgments on the correctness of answers tend to be consistent.**  We sampled 20 GPT-4o prediction samples on StreamBench for verification and found that the judgment results of LLaMA-3 were slightly different from those of humans.
> >
> > The experimental data is shown in the following table:
> >
> > | Judge    | Sco. | Acc. |
> > |--|--|--|
> > | Human     | 2.55 | 65.0 |
> > | LLaMA-3 | 3.15 | 65.0 |
> >
> > LLaMA-3 tends to give higher scores, **but is on par with human judgement in terms of accuracy.** We have also provided some examples for your convenience:
> >
> > Same score:
> > ```json
> > {"id": 180, "question": "I can't quite remember, who used the knife to cut the beef?", "answer": "It was the man in the white shirt.", "class": "LM", "pred": "I can't tell who used the knife in the images since I can't identify or describe faces. However, it looks like several cooking activities involving a knife are shown, with people in chef uniforms working together.", "llama_pred": "no", "llama_score": 2, "human_pred": "no", "human_score": 2}
> >
> > {"id": 120, "question": "Is there any scissors in this video?", "answer": "Yes, there is a pair of scissors above the silver pot.", "class": "CS", "pred": "Yes, there are scissors visible in several parts of the video, hanging on the wall on a kitchen rail behind the person cooking.", "llama_pred": "yes", "llama_score": 5, "human_pred": "yes", "human_score":3 }
> >
> > {"id": 120, "question": "Is there any scissors in this video?", "answer": "Yes, there is a pair of scissors above the silver pot.", "class": "CS", "pred": "Yes, there are scissors visible in several parts of the video, hanging on the wall on a kitchen rail behind the person cooking.", "llama_pred": "yes", "llama_score": 5, "human_pred": "yes", "human_score":3 }
> >
> > {"id": 120, "question": "Where is a glass or bowl containing red, thick food with a spoon in it?", "answer": "It may be on the cupboard near the cutting board or it is at corner of the cutting board, next to the woman.", "class": "FT", "pred": "The glass or bowl containing red, thick food with a spoon in it is on the countertop, near where the woman is working on the cutting board.", "llama_pred": "yes", "llama_score": 5, "human_pred": "yes", "human_score":5}
> >
> > ```
> >
> > Different score:
> > ```json
> > {"id": 54, "question": "Can you tell me where in the current room I can find the red fire extinguisher?", "answer": "Yes, it looks like there is a red fire extinguisher mounted on the wall near the refrigerator.", "class": "SM", "pred": "The red fire extinguisher is located on the wall in the kitchen area, near the refrigerator.", "llama_pred": "yes", "llama_score": 5, "human_pred": "yes", "human_score": 4}
> >
> > {"id": 88, "question": "Where did I place the cup that I just filled with coffee grounds?", "answer": "If I\u2019m not mistaken, you placed the cup on the countertop, next to a clear plastic container and a stainless steel pot lid.", "class": "LM", "pred": "The cup filled with coffee grounds was placed on the countertop next to a dish drainer and a moka pot lid.", "llama_pred": "yes", "llama_score": 4, "human_pred": "yes", "human_score":3}
> >
> > {"id": 220, "question": "Based on what you see, Did the man in black clothing handle a green shovel?", "answer": "Yes, the man in black handled a green shovel in his hand", "class": "LM", "pred": "Based on the images provided, there is no indication that the man in black clothing handled a green shovel.", "llama_pred": "no", "llama_score": 0, "human_pred": "no", "human_score":2 }
> >
> > {"id": 88, "question": "Where did I place the cup that I just filled with coffee grounds?", "answer": "If I\u2019m not mistaken, you placed the cup on the countertop, next to a clear plastic container and a stainless steel pot lid.", "class": "LM", "pred": "The cup filled with coffee grounds was placed on the countertop next to a dish drainer and a moka pot lid.", "llama_pred": "yes", "llama_score": 4, "human_pred": "yes", "human_score":3}
> >
> > ```

---

> > > ### Author Response · Authors · 2024-11-25
> > > **Official Comment by Authors**
> > >
> > > **Q2: The motivation behind architectural design is not clear enough.**
> > >
> > > **A2: We fully understand your and 9hDR's concerns about the motivation for memory design and we have therefore reorganized the presentation of Section 3 (L212-L323) in the revised version.**
> > >
> > > **First of all, our method regards the video as a dynamic information repository, and we hope to find a human-like memory method that can extract the most relevant knowledge from the information based on the user's answer as a contextual supplement to answer the question.**
> > > The first challenge we need to face is to reduce the feature storage overhead caused by redundant frames in videos.
> > > To overcome this challenge, we design a selective frame stacking module to remove the redundant one in consecutive video frames.
> > > By calculating the momentum vector between adjacent video frames, the dynamic change of the current video frame relative to the past moment can be determined, thereby deciding whether to retain the video frame.
> > >
> > > **Second, inspired by the Atkinson-Shiffrin model, we divide memory into long-term and short-term.** Short-term memory retains clear, specific info briefly and updates often. E.g., when watching videos, we usually focus on current events rather than each frame's details. So, $M_s$ stores vision features StreamChat's short-term memory. As humans, while watching a movie, we need to recall detailed info to fill short-term memory gaps, which prompts long-term memory.
> > >
> > > We believe that the main conflict point is how to find relevant information in the memory tree. **Here, text clues (introduced in Section 3.1 L262-L266) serve as an index for retrieving relevant information from the long-term memory.**
> > > **As introduced in Section 3.1.3 L305-309, by calculating the cosine similarity between the question and text clues ,** we can locate the most relevant memory information in Ml and extract it. At the same time, we provide a transparent reasoning process case in Figure 6, hoping to help more researchers understand our method.
> > >
> > > **Third, to improve the interactive experience in multi-round dialogue scenarios,** we save the historical conversation records between users and StreamChat as dialogue memory, so that StreamChat can generate more targeted answers. Just like we introduced in Figure 6, the retrieved record will be supplemented as new contextual information to help the model answer the correct answer.
> > >
> > > **Q3: This setting seems to be specially optimized to deploy the authors' model, which makes the efficiency comparison with other models unfair.**
> > >
> > > **A3: We fully understand your concerns.** In this paper, we are trying to find a way to improve video-llms to streaming video understanding agent, **a system-level solution by decoupling the components.**
> > > According to the reviewers’ comments, we validated that our approach can be extended to more video-llm, even though they have not been previously trained on specialized data.
> > > **We update the motivations of model selection in the Appendix F (L966-L1005).** Therefore, we want to prove that our method is a generally applicable method, not just for a specific model. If you have any questions feel free to ask.

---

### Official Review · Reviewer_Bg29 · 2024-11-03

**Soundness:** 3
**Presentation:** 3
**Contribution:** 1
**Rating:** 6
**Confidence:** 4

**Summary:**

In this work, the authors propose a training-free framework, StreamChat, for streaming video understanding. StreamChat uses a hierarchical memory system to efficiently process long video sequences for real-time, multi-turn dialogue. With parallel scheduling, it improves speed and reduces latency.

The authors also introduce a new benchmark, StreamBench, which tests video understanding across diverse scenarios, and results show StreamChat outperforms current models in accuracy and response time.

**Strengths:**

S1: The paper introduces a novel hierarchical memory mechanism that compresses video representations over long sequences, enabling efficient video feature retrieval in real-time multi-turn dialogue contexts.

S2: I believe the introduction of StreamBench as a benchmark for streaming video understanding is a significant contribution that can fill a critical gap in the field. By offering a standardized evaluation framework, StreamBench enables more rigorous comparisons across models, driving advancements in streaming video understanding.

S3: The proposed StreamChat achieves the best performance on StreamBench, surpassing other state-of-the-art methods, which is impressive.

**Weaknesses:**

W1: StreamChat was compared only with several open-source models. Including proprietary models like GPT-4o and Gemini-1.5 in the StreamBench evaluation would provide a more comprehensive comparison.

W2: Human performance is absent in the comparison, which would help illustrate the gap between current models and human capabilities if included.

W3: The authors compare their new benchmark only with older datasets or benchmarks, such as MSVD, MSRVTT, and ActivityNet. It would strengthen the evaluation to include comparisons with more recent benchmarks for video understanding, such as Seed-Bench [1], Video-Bench [2], MVBench [3], and LVBench [4].

W4: The paper lacks an analysis of factors that impact performance on StreamBench, such as the input frame sequence length or the language model size used by the models.

[1] Li, Bohao, et al. "Seed-bench: Benchmarking multimodal llms with generative comprehension." arXiv preprint arXiv:2307.16125 (2023).

[2] Ning, Munan, et al. "Video-bench: A comprehensive benchmark and toolkit for evaluating video-based large language models." arXiv preprint arXiv:2311.16103 (2023).

[3] Li, Kunchang, et al. "Mvbench: A comprehensive multi-modal video understanding benchmark." Proceedings of the IEEE/CVF Conference on Computer Vision and Pattern Recognition. 2024.

[4] Wang, Weihan, et al. "LVBench: An Extreme Long Video Understanding Benchmark." arXiv preprint arXiv:2406.08035 (2024).

**Questions:**

Q1: Will the example in the prompt impact the models' prediction? For example, your example in the prompt is "{'pred': 'A'}". Does it increase the probability of predicting "A"?

Q2: The StreamBench benchmark supports only QA tasks now. Are there any plans to extend it to include additional streaming video understanding tasks, such as dense streaming video captioning or temporal/spatiotemporal grounding?

---

> ### Author Response · Authors · 2024-11-23
> **Official Comment by Authors**
>
> **Thank you for your insightful feedback. We appreciate your recognition of the efficiency of our hierarchical memory storage design and the contribution of StreamBench. Our detailed responses to your concerns are provided below:**
>
> **Q1: Including models like GPT-4o and Gemini-1.5 in the StreamBench evaluation.**
>
> **A1: Tank you for your suggestion on performance of GPT-4o [1] and Gemini-1.5 [2] in the StreamBench.Adding the performance results can make our work more comperhensive.**
> To resolve your doubts, we added their results to StreamBench and made corrections in the paper.
> The results are shown in the following table.
>
> | Method | FPS | Fr. | Sco. | Acc. | Coh. |
> | ---- | ---- | ---- | ---- | ---- | ---- |
> | GPT-4o      | -- | 50 | 3.70 | 71.0 | 1.66 |
> | GPT-4o      | -- | 35 | 3.64 | 69.8 | 1.72 |
> | GPT-4o-mini | -- | 35 | 3.17 | 59.1 | 2.01 |
> | StreamChat-Slow    | 15 | -- | 3.48 | 64.7 | 1.76 | 0.90 |
> | StreamChat-Base    | 20 | -- | 3.42 | 63.8 | 1.79 | 0.89 |
> | StreamChat-Fast    | 32 | -- | 3.28 | 61.7 | 1.81 | 0.85 |
>
>
> | ~ | OS | LM | SM | CI | KG | SF |
> | --- | --- | --- | --- | --- | --- | --- |
> | Method      | Sco./Acc. | Sco./Acc. | Sco./Acc. | Sco./Acc. | Sco./Acc. | Sco./Acc. |
> | GPT-4o_50      | 3.27/60.5 | 3.35/61.2 | 3.41/64.4 | 3.81/72.3 | 4.58/93.9 | 3.83/74.7 |
> | GPT-4o_35      | 3.22/59.6 | 3.28/58.6 | 3.45/65.3 | 3.76/71.7 | 4.54/93.3 | 3.50/66.1 |
> | GPT-4o-mini_35 | 2.52/46.8 | 2.70/45.8 | 2.80/51.0 | 3.50/64.0 | 4.67/95.2 | 2.90/53.3 |
>
> In order to evaluate the actual performance of GPT-4o and GPT-4o-mini, we sampled 50 frames and 35 frames respectively, which is the **maximum number of frames** that we found in our experiments to be able to obtain **stable inference results.**
>
> According to our experimental results, we find that **GPT-4o using 50 frames outperforms our method by 6.3\%**, and **GPT-4o-mini using 35 frames performs 5.6\% lower than ours**.
> To obtain more comperhensive comparision, we conduct experiment of GPT-4o with 35 frames.
> The results show that the performance of **GPT-4o with 35 frames input will decrease slightly compared to sampling 50 frames, but it is still 5.1\% higher than our method**.
>
> We supplemented the experimental results in Tables 4 and 5 in Section 4 of the paper.
>
> **Q2: Human performance is absent in the comparison.**
>
> **A2: Thank you for your suggestion. The performance of humans on the Benchmark can make our work more complete and reasonable.** To this end, we specially found four researchers with master's degrees to conduct human evaluation. We added the experimental results to the benchmark and made corrections in the paper.
>
> | Method | FPS | Fr. | Sco. | Acc. | Coh. |
> | ---- | ---- | ---- | ---- | ---- | ---- |
> | Human_performance  | -- | -- | **4.03** | **79.4** | **1.16** |
> | GPT-4o      | -- | 50 | 3.70 | 71.0 | 1.66 |
>
>
> | ~ | OS | LM | SM | CI | KG | SF |
> | --- | --- | --- | --- | --- | --- | --- |
> | Human_performance  | **3.95/71.8** | **3.81/69.3** | **4.07/81.5** | **4.14/82.6** | 4.06/80.7 | **4.32/92.5** |
> | GPT-4o_50          | 3.27/60.5 | 3.35/61.2 | 3.41/64.4 | 3.81/72.3 | **4.58/93.9** | 3.83/74.7 |
>
> According to our results, humans outperform GPT-4o on most of tasks, but GPT-4o outperforms humans on knowledge-based tasks (without the help of external tools). **This may be because GPT-4o's huge pre-training data and memory ability allow it to face knowledge-based questions more easily,** while humans need the help of search engines to get more answers to questions.
>
> We supplemented the experimental results in Tables 4 (L350-367) and 5 (L380-397) in Section 4 of the revised manuscript.

---

> > ### Author Response · Authors · 2024-11-23
> > **Official Comment by Authors**
> >
> > **Q3: Include comparisons with more recent benchmarks for video understanding, such as Seed-Bench, Video-Bench, MVBench, and LVBench.**
> >
> > **A3:**
> > Thank you for the suggestion to include additional benchmarks. We have investigated evaluating StreamChat on
> > SEED-Bench-2 [3], Video-Bench [4], MVBench [5], and LVBench [6].
> >
> > On SEED-Bench-2, the limited number of frames (4-8) per video hinders StreamChat's ability to leverage its temporal modeling capabilities, which are designed for streaming video.  Consequently, our performance is completely consistent  with the LongVA baseline on this benchmark, as shown below.
> >
> > | Method  | Language Model | Video | Part-1 | Part-2 | Part-3 |
> > | ---- | ---- | ---- | ---- | ---- | ---- |
> > | ---- | ---- | AvgAcc. | AvgAcc. | AvgAcc. | AvgAcc. |
> > | Stream-Chat-Base    | Qwen2-7B        | 27.2 | 28.2 | 21.3 | 0 |
> > | LongVA              | Qwen2-7B        | 27.2 | 28.2 | 21.3 | 0 |
> >
> > Regarding the poor performance of our method on SEED-Bench, our analysis results are as follows:
> >
> > **First, since our method pays more attention to the processing ability under lengthy videos, and SEED-Bench only provides us with 4-8 frames of video information,** as our method mainly focuses on video streaming scenarios, this is not conducive to our method. We will further improve our method to adapt to more scenarios and user needs.
> >
> > **Second, since the setting of SEED-Bench, we removed all the memory storage and evaluated the performance of LongVA.** We found that LongVA also exhibited suboptimal performance. We will continue to explore  methods to make up for this defect.
> >
> > We also made efforts to evaluate our method on Video-Bench, MVBench, and LVBench, but encountered several obstacles.  **We successfully ran experiments on Video-Bench, but the submission website malfunctioned, preventing us from retrieving our scores. This [issue](https://github.com/PKU-YuanGroup/Video-Bench/issues/8) appears to be affecting other researchers as well.**
> >
> > For MVBench, copyright restrictions on some videos hindered our evaluation.  **We have applied for the necessary usage rights but have not yet received approval.**  Similarly, access to the YouTube data used in LVBench requires approval, which we have not yet received despite application. **We are actively working to address these issues and will provide results on these benchmarks if access becomes available.**
> >
> > **Q4: The paper lacks an analysis of factors that impact performance on StreamBench.**
> >
> > **A4: We appreciate your feedback regarding the analysis of factors impacting performance on StreamBench.**  We address the effect of input sequence length and language model size below.
> >
> > **First, regarding input sequence length, Section 4.4 (Design of Long Memory Tree, Figure 7(c)) and the table below explores this impact.**  $C$ (clustering goal) represents the length of each clustered visual feature, which is related to the input sequence length.
> > Based on our memory design, the final input vision token length is equal to $C$ x $L$.
> > The results show that increasing the sequence length generally improves performance (Scor, Acc, Cho, RDP) but also increases memory consumption.  The performance gains diminish with increasing sequence length.  To balance performance and resource consumption, we chose a sequence length of 5 (C=5) for our experiments.
> >
> > | $L$ | $g$ | $C$ | Scor | Acc | Cho | RDP | Memory |
> > | --- | --- | --- | --- | --- | --- | --- | --- |
> > | 25 | 10 | 2 | 3.27 | 59.4 | 1.91 | 0.87 | 20 |
> > | 25 | 10 | 3 | 3.35 | 62.1 | 1.83 | 0.88 | 24 |
> > | 25 | 10 | 5 | 3.42 | 63.8 | 1.79 | 0.89 | 32 |
> > | 25 | 10 | 8 | 3.42 | 63.9 | 1.79 | 0.92 | 39 |
> > | 25 | 10 | 10 | 3.43 | 64 | 1.78 | 0.92 | 46 |
> > | 25 | 10 | 12 | 3.43 | 64.1 | 1.78 | 0.93 | 53 |
> > | 25 | 10 | 15 | 3.43 | 64.2 | 1.77 | 0.91 | 61 |
> > | 25 | 10 | 20 | 3.44 | 64.2 | 1.77 | 0.89 | 67 |
> >
> > We appreciate your suggestion and have reorganized the description of the experimental setup in Section 4.4 to make clear the impact of the input sequence on model performance.
> >
> > **Second,** regarding language model size, we agree that larger models generally lead to improved performance, as demonstrated by scaling laws.  **However, LongVA provieded us the 7B model and we focused on models with 8B parameters or less to maintain comparable computational requirements and ensure a fair comparison.**  Exploring larger models with increased data and computational resources is a promising direction for future work.

---

> > > ### Author Response · Authors · 2024-11-23
> > > **Official Comment by Authors**
> > >
> > > **Q5: Will the example in the prompt impact the models' prediction?**
> > >
> > > **A5: If I understand correctly, the prompt {'pred': 'A'}  you are referring to should appear in the Appendix A(L737-L743).** We are sorry that our statement misled you. **The prompt here is actually used in the data collection pipeline to help VLM achieve accurate classification of videos.** Therefore, the prompt here has no effect on our experimental results. We will correct these statement issues in the manuscript (see Appendix A (L737-L743) ).
> > >
> > > **Q6: Plans to extend StreamBench to include more streaming video understanding tasks.**
> > >
> > > **A6: Definitely.** it is in our future pan to extend the scale of StreamBench and tasks that it contains.
> > > We plan to expand the benchmark data from three aspects:
> > >
> > > **First, more tasks.** Including more tasks can further improve the diversity of the benchmark and test the capabilities of the model in more aspects. We will add multiple-choice questions, grounding, and captioning tasks to further improve the completeness of the benchmark.
> > >
> > > **Second, more video source.** Although our benchmark contains videos carefully selected from EgoSchema [7] and Youtube-8M [8], the types of videos we cover are still limited. We plan to add more types of data, such as sports, short videos, etc. to enrich streambench.
> > >
> > > **Third, more modalities.** The information covered by the streaming video is very extensive, like audio signals are also included. We will incorporate more modalities test data into the benchmark to provide a wider range of test results.
> > >
> > > [1] Gpt-4o system card. arXiv 2024.10
> > >
> > > [2] Gemini 1.5: Unlocking multimodal understanding across millions of tokens of context. arXiv 2024.03
> > >
> > > [3] SEED-Bench-2: Benchmarking Multimodal Large Language Models. CVPR 2024
> > >
> > > [4] Video-Bench: A Comprehensive Benchmark and Toolkit for Evaluating Video-based Large Language Models.   arXiv 2023.11
> > >
> > > [5] MVbench: A comprehensive multi-modal video understanding benchmark. CVPR 2024
> > >
> > > [6] LVbench: An extreme long video understanding benchmark. arXiv 2024.06
> > >
> > > [7] Egoschema: A diagnostic benchmark for very long-form video language understanding. NeurIPS 2023
> > >
> > > [8] Youtube-8m: A large-scale video classification benchmark. arXiv 2016.09

---

> ### Comment · Reviewer_Bg29 · 2024-12-01
>
> Thank you very much for your response. All my concerns have been addressed.
>
> I firmly believe that streaming video understanding is a highly promising and impactful area of research with potential far beyond QA tasks, so I'm looking forward to your expanding StreamBench to encompass a broader range of streaming video understanding tasks.

---

### Official Review · Reviewer_qLQa · 2024-11-04

**Soundness:** 3
**Presentation:** 2
**Contribution:** 3
**Rating:** 6
**Confidence:** 3

**Summary:**

This paper presents a benchmark, named StreamBench, and as well a model, named StreamChat, for real-time streaming video LLM answering. The StreamBench comprehensively evaluates assorted capabilities of video LLMs, including spatial search, long & short memory recall, common knowledge reasoning, etc. The size of StreamBench is also larger than previously-proposed benchmarks. The authors also paid a lot of efforts in building the training-free StreamChat model, which bases on current video LLM LongVA and consists of novel techniques including hierarchical memory storage, optimized system scheduling. Evaluations on various benchmarks show that StreamChat enjoys better performance with lower latency compared to previous streaming models.

**Strengths:**

1. The proposed benchmark, StreamBench, is a comprehensive benchmark testing various capabilities of real-time video LLMs. As far as I know, this is the first benchmark designed for online streaming video task evaluation.
2. The proposed method, StreamChat, integrates frame selection, memory building and retrieval, and several optimizations into a video LLM in a training-free manner. Though designed complicatedly, the latency of StreamChat is still under control (under one second). Several benchmarks, including online StreamBench and offline video understanding tasks, show the effectiveness of the proposed StreamChat. The authors also put a detailed inference example, every components illustrated, in Fig 6. Also, the detailed ablations on threshold and memory parameters should be appreciated.

**Weaknesses:**

My major concerns comes from two sides:
1. **The presentation**. The presentation of StreamBench is good but the part of StreamChat seems to need to be further improved. It is hard to tell what StreamBench is designed to do when understanding Fig 4 and Fig 5. Though there are both captions and texts written in the method section, it is still hard to relate text descriptions to components in the figures (especially for ones less familar with memory mechanism and RAG).
2. **The complexity**. StreamChat seems too complicated: there are selective frame stacking based on motion vectors, vision-based short-term memory, caption-based long-term memory (also a clustering mechanism in it), dialogue memory (whose language encoder is different from video LLM), and FAISS indexing library. Although being tuning-free, all these components make this 'system' super heavy. The authors should justify every components used in building the memory system in StreamChat. Moreover, this looks more like a RAG system specifically designed for online videos. The authors should compare with other basic RAG methods.

**Questions:**

See weaknesses. I am just at the borderline due to 1) the contribution of StreamBench and StreamChat, 2) the over-complicated design of StreamChat (being too engineering and RAG-like).

---

> ### Author Response · Authors · 2024-11-23
> **Official Comment by Authors**
>
> **Thank you for your insightful feedback. We appreciate your recognition of the novelty of StreamBench and the extensive experiments and analysis conducted for StreamChat. Our detailed responses to your concerns are provided below:**
>
> **Q1: The presentation. The part of StreamChat seems to need to be further improved.**
>
> **A1: We appreciate the reviewer's feedback on the presentation of StreamChat.** We recognize the importance of clear and accessible presentation and have taken steps to improve this aspect in the revised manuscript. **Specifically, we have rewritten Section 3 (L 254-267) to provide a more detailed and clearer explanation of StreamChat's architectural design.**   Furthermore, we have revised **Figures 4 and 5,** adjusting the colors and overall structure to enhance readability and highlight key components and relationships.  We believe these changes will significantly improve the understanding of StreamChat.
>
> **Q2: The contribution of StreamChat.**
>
> **A2:** We want to find resource-friendly architecture to help video-llms achieve understanding of streaming videos, including real-time video processing , management of video information and multi-round interaction.
> For that purpose, we design and complete StreamChat, an training-free streaming video agent.
> **Different from existing video-llms that treat videos as simple frames and features, StreamChat views videos as dynamic information repositories and uses hierarchical memory to parse and store the various information contained in the video,** including vision features and captions describing what is happening in the video. When encounter user requests, StreamChat will search for the most related vision features in memory as relevant tokens to help model answer correctly.
>
> It's like a "video version" RAG. **However, there are two distinct differences between StreamChat and RAG system.**
>
> **First, different memory knowledge sources.** Most RAG systems rely on regularly updated text libraries or external knowledge bases to provide the latest information in real time through retrieval[1]. StreamChat only relies on videos that we feed to it , and it can extract and update information from videos itself.
>
> **Second, less resource consumption makes it more practical.** The RAG retrieval process and generation process each require computing resources, especially when querying large document or multi-modal repositories [2]. As our StreamChat using K-Means algorithm to compress vision features, it can be imployed within limited resource.
> We are sorry that our complex paper presentation hindered your understanding of the StreamChat structure and the refined version is well organized.
>
> We appreciate your suggestion and we will further emphasize the differences between our method and RAG in the revised version so that everyone can better understand our research.

---

> > ### Author Response · Authors · 2024-11-23
> > **Official Comment by Authors**
> >
> > **Q3: StreamChat seems too complicated. The authors should justify every components used in building the memory system in StreamChat.**
> >
> > **A3:** In order to make StreamChat more adaptive to dynamic stream video scenarios, we have designed vision-based short-term memory, caption-based long-term memory and dialogue memory. **However, in order to make these components work smoothly together, we also designed system scheduling**, which decouples the operation of each component into different threads, thus ensuring low latency and real-time video processing.
> >
> > Our intention was to design a human-like memory method that simulates the Atkinson-Shiffrin model[3].
> > The design of short-term memory is to focus on what is happening right now. Just as we described in **Section 3.1.1** , we utilize the continuously updated buffer to simulate human short-term memory.
> > The long-memory tree simulates the complex and abstract memory of human. We use a clustering algorithm to perform secondary compression on visual features to simulate the fragmented storage of human memories of past events, while also reducing the required storage space. In the meantime, we use captions of each video clip as the index to search for the most related information.
> > **Our experiment results in Section 4.4 Table 7** verifies the effectiveness of each components.
> >
> > | $M_l$ | $M_s$ | $M_d$ | OS | LM | SM | CI | KG | SS | Average |
> > | --- | --- | --- | --- | --- | --- | --- | --- | --- | --- |
> > | --- | --- | --- | Sco./Acc. | Sco./Acc. | Sco./Acc. | Sco./Acc. | Sco./Acc. | Sco./Acc. | Sco./Acc. |
> > |          |          |          | 2.54/41.6 | 2.55/45.5 | 2.93/52.5 | 3.30/60.1 | 4.44/89.9 | 3.79/72.6 | 3.27/60.3 |
> > |          |          | &#10004; | 2.55/41.9 | 2.55/45.7 | 2.94/52.5 | 3.66/64.2 | 4.44/88.7 | 3.78/72.4 | 3.32/60.9 |
> > |          | &#10004; |          | 2.58/43.3 | 2.62/46.6 | 3.09/55.7 | 3.31/60.7 | 4.39/88.1 | 3.68/69.8 | 3.28/60.7 |
> > | &#10004; |          |          | 2.85/49.5 | 2.78/51.7 | 2.96/53.5 | 3.32/61.1 | 4.42/88.4 | 3.65/69.4 | 3.33/62.2 |
> > | &#10004; | &#10004; |          | 2.91/50.4 | 2.88/53.0 | 3.10/56.0 | 3.55/63.4 | 4.36/87.6 | 3.58/68.7 | 3.39/63.1 |
> > | &#10004; | &#10004; | &#10004; | 2.93/50.5 | 2.87/52.9 | 3.15/56.1 | 3.82/67.6 | 4.37/87.9 | 3.56/68.8 | 3.42/63.8 |
> >
> > According to our experiments, adding long-term memory $M_l$ improved the LM task performance by 6.2\%, while the use of short-term memory $M_s$ boosts SM task performance by 3.2\%.
> > The results indicate that the model’s performance in each subtask aligns with the inclusion of specific memory attributes.
> > Additionally, we observe that different memory components can complement each other. When both long-term $M_l$ and short-term memory $M_s$ are applied simultaneously, the average accuracy increases by 0.9\%.
> >
> > **Q4: Looks more like a RAG system specifically designed for online videos. Authors should compare with more RAG system.**
> >
> > **A4: We appreciate the reviewer's suggestion to compare our method with more Retrieval Augmented Generation (RAG) systems.**  We agree that this comparison would be valuable. We are aware of recent work [4, 5] applying RAG to video understanding and addressing hallucination issues. However, these methods currently lack publicly available code, precluding direct experimental comparison.
> >
> > We are committed to incorporating these comparisons as soon as the code becomes available, providing a more comprehensive analysis of different approaches on StreamBench.
> > Furthermore, we understand the reviewer's implicit concern about the distinction between our approach and RAG systems. We will revise the paper to more clearly articulate the key differences in motivation and operational mechanisms.
> >
> > [1] Retrieval-augmented generation for ai-generated content. arXiv 2024.02
> >
> > [2] Memorybank: Enhancing large language models with long-term memory. AAAI 2024
> >
> > [3] A proposed system and its control processes. _The Psychology of Learning
> > and Motivation_ 1968
> >
> > [4] iRAG: Advancing RAG for Videos with an Incremental Approach. CKIM 2024
> >
> > [5] ViTA: An Efficient Video-to-Text Algorithm using VLM for RAG-based Video Analysis System. CVPR 2024

---

> > > ### Comment · Reviewer_qLQa · 2024-12-02
> > >
> > > Than you for your detailed response. I have read comments by other reviewers as well as authors' responses to them and my concerns have been properly addressed. Now I am leaning to a borderline accept.

---

### Official Review · Reviewer_9hDR · 2024-11-07

**Soundness:** 3
**Presentation:** 3
**Contribution:** 3
**Rating:** 6
**Confidence:** 3

**Summary:**

This paper introduces a method for online interactions with videos through an LLM, and a benchmark suitable for evaluating streaming-style scenarios where the dialogue happens in real time with the video.

Contributions:
- Video-based LLM which processes high framerate video (typically LLMs need to subsample severely the video due to computational constraints). This method uses optical flow magnitude to assess whether to process any particular frame (w.r.t. previous), therefore saving considerable amount of computation needed. The LLM also has a "short/long" term memory system.
- New benchmark that should be useful since it's one of a kind, simulating multi-round interactions centered on the video content. 25h of content was used in the creation of this benchmarks, making it one of the larger ones available.

**Strengths:**

Strong empirical results. This is a systems pape which is tied to a particular benchmark which was designed for it, so it's unsurprising that the proposed method works best. With that said, given the engineering effort put into it, it still is impressive, particularly because of the fact that this system could be used in a realtime scenario.

I am personally excited by the new benchmark that is proposed in this paper perhaps even more so than the method, because it captures a whole class of interactions which were under-represented before.

**Weaknesses:**

I am kind of baffled by the language of "our method can further improve our foundation model LongVA [20] by 5.1% in accuracy". Does this imply that you're the authors of this method? (Please don't answer this question, but please be mindful of the language which might accidentally de-anonimize you).

The hierarchical design seems a bit ad-hoc. It would have been nice to spend a bit more time to motivate the various design choices. Currently, it sounds like the paper is in the "this is what we did" style, without spending too much into the motivation which led to the various choices made. I could easily imagine many other ways in which the hierarchical memory could have been implemented.

The abstract claims "compression" of video features, but reading the paper reveals that this is actually not at all the case. I find the abstract to be misleading and should be updated to reflect the reality of the methods involved.

Minor comments:
- I am having a hard time reading figures. For example, in Figure 1 I had to use a color picker to figure out which method was which line. In the interest of color blind people, (or partially color blind people like me), please use symbols (*, triangle, diamond, etc) or line styles in addition to colors to make the figures more accessible (either to color-challenged people like me, or when printed out in grayscale).
- There's a typo "Gine-grained" in the appendix.

**Questions:**

- Do you plan to make this benchmark publicly available?
- Do you plan to open source your method so that others may benefit from the engineering effort you've put into your method? (I am of course talking about academic researchers)
- What the process of identifying the various categories STREAMBENCH encompasses? I am of course talking about the domains and classes of videos.
- In the appendix, you very briefly mention that you're using a LLM to determine accuracy. The formula you present is unclear. (What's N, T?). What are you getting out of the LLaMA model? Also, do you use any particular prompt for the LLaMA model?
-

---

> ### Author Response · Authors · 2024-11-23
> **Official Comment by Authors**
>
> **Thank you for your insightful feedback. We appreciate your positive feedback on StreamBench's interactive features and memory storage design. Our detailed responses to your concerns are provided below:**
>
> **Q1: Motivations of compients' design.**
>
> **A1:** We reorganize the description statements to help you understand the design motivations of each component more intuitively in our hierarchical design.
>
> **First, to reduce the feature storage overhead caused by redundant frames in videos, we design a selective frame stacking module to remove the redundant one in consecutive video frames.** As shown in Section 3 Figure 4, there is no obvious change in the content and character movements between the third and fourth pictures, so the momentum vector calculated by the optical flow algorithm will be very small. Therefore, removing the fourth picture does not affect the model's understanding of the global or partial scene.
>
> **Second, our memory storage simulates human memory patterns.** Inspired by the Atkinson-Shiffrin model[2], we split memory into long-term memory and short-term memory. The short-term memory  in preserving clear and specific information for a brief duration with frequent updates, just like us. When watching videos, most of the time we mainly focus on what is happening right now rather than every detail of each frames. So we utilize Ms to store these vision features as the short-term memory of StreamChat. However, as humans, we still need to remember detailed information while watching a movie to make up for the missing details in short-term memory and that is the motivation of long-term memory. To overcome the bottleneck of VRAM consumption and the effectiveness of retrieving memory, we design a tree structure as shown in Section 3 Figure 5.
>
> **Third, to improve the interactive experience in multi-round dialogue scenarios,** we save the historical conversation records between users and StreamChat as dialogue memory, so that StreamChat can generate more targeted answers based on historical records.
> We hope that these descriptions can resolve your doubts, and we will also make changes to the revised version of the manuscript.
>
> Based on your suggestions, we refine our presentation of the motivations for the design of each component in the revised version of the manuscript in Section 3 (L254-L267).
>
> **Q2: The abstract claims "compression" of video features, but reading the paper reveals that this is actually not at all the case.**
>
> **A2: We apologize that our description may have misled you, however, we do carry out a compression process in our memory storage.**
> Even though the selective frame stacking module has already helped us remove redundant frames, the remaining video features are lengthy: 256 tokens each frame. **In this case, we use a K-Means algorithm to compress features into shorter lengths as we introduced in Secton 3.1.1 part: Long-term Memory.** In our setting, the original feature length is 256 and we compress it to 5, which significantly reduce the VRAM consumption.
>
> The experiment resutls in **Section 4.4, part:Design of Long Memory Tree, Figure 7(c)** shows the impact of compression on performance. We present all the data in the following table.
>
> | $C$ | Scor | Acc | Cho | RDP | Mem |
> | --- | --- | --- | --- | --- | --- |
> | 2 | 3.27 | 59.4 | 1.91 | 0.87 | 20 |
> | 3 | 3.35 | 62.1 | 1.83 | 0.88 | 24 |
> | 5 | 3.42 | 63.8 | 1.79 | 0.89 | 32 |
> | 8 | 3.42 | 63.9 | 1.79 | 0.92 | 39 |
> | 10 | 3.43 | 64.0 | 1.78 | 0.92 | 46 |
> | 12 | 3.43 | 64.1 | 1.78 | 0.93 | 53 |
> | 15 | 3.43 | 64.2 | 1.77 | 0.91 | 61 |
> | 20 | 3.44 | 64.2 | 1.77 | 0.89 | 67 |
>
> The $C$(clustering goal) represents the final vision token length that we stored in the long-term memory tree.
> **The data shows that increasing $C$ will significantly increase the memory consumption, but the performance improvement is limited.** For this reason, we chose a trade-off between performance and memory and used $C$=5 as the standard parameter.
>
> **We improved the statement in Section 4.4  (L472-L473) in the revised version of the paper** to help everyone better understand the impact of compression on our method.
>
> **Q3: Having a hard time reading figures.**
>
> **A3: We appreciate you bringing the figure readability to our attention.**  In the revised manuscript, we have improved Figure 1  by incorporating distinct symbols (e.g., circles, triangles, squares), ensuring clear differentiation of the methods. We have also reviewed and similarly improved all other figures to enhance overall readability.

---

> > ### Author Response · Authors · 2024-11-23
> > **Official Comment by Authors**
> >
> > **Q4: Do you plan to make this benchmark publicly available?**
> >
> > **A4: Yes, we plan to release our code and benchmark data publicly to benefit the research community.**
> > We are currently preparing the code and benchmark for release and will make it available on github as soon as possible.
> >
> > **Q5: What the process of identifying the various categories SteramBench encompasses?**
> >
> > **A5: The details of our data collection pipeline in Appendix A (L718-L750) may help you solve this question.**
> > Our data sources are EgoSchema[3] and Yotube-8M[4]. The video information from Yotube-8M already comes with the category label corresponding to each video when it is downloaded, so we can use it directly. The videos from EgoSchema do not come with category labels, but the paper has a total category set of all videos, so we use Video-LLaVA[5] to help us determine the category of each ego video.
> >
> > **Q6: Carification on the LLM-based accuracy calculation described in the appendix, asking about the formula's variables, the LLaMA model's output, and the specific prompt used.**
> >
> > **A6: We sincerely sorry for our loss.** Prompts that are used for the LLaMA-3[6] model can be a very important part of getting right results. The following is the prompt that we use:
> > ```json
> > {
> >     "role": "system",
> >     "content":
> >     "You are an intelligent chatbot designed for evaluating the correctness of generative outputs for question-answer pairs. "
> >     "Your task is to compare the predicted answer with the correct answer and determine if they match meaningfully. Here's how you can accomplish the task:"
> >     "------"
> >     "##INSTRUCTIONS: "
> >     "- Focus on the meaningful match between the predicted answer and the correct answer.\n"
> >     "- Consider synonyms or paraphrases as valid matches.\n"
> >     "- Evaluate the correctness of the prediction compared to the answer."
> >     },
> >     {
> >     "role": "user",
> >     "content":
> >     "Please evaluate the following video-based question-answer pair:\n\n"
> >     f"Question: {question}\n"
> >     f"Correct Answer: {answer}\n"
> >     f"Predicted Answer: {prediction}\n\n"
> >     "Provide your evaluation only as a yes/no and score where the score is an integer value between 0 and 5, with 5 indicating the highest meaningful match. "
> >     "Please generate the response in the form of a Python dictionary string with keys 'llama_pred' and 'score', where value of 'llama_pred' is  a string of 'yes' or 'no' and value of 'score' is in INTEGER, not STRING."
> >     "DO NOT PROVIDE ANY OTHER OUTPUT TEXT OR EXPLANATION. Only provide the Python dictionary string. "
> >     "For example, your response should look like this: {'llama_pred': YOUR_JUDGE, 'score': YOUR_SCORE}."
> > }
> > ```
> >
> > After we feed this prompt to LLaMA-3, we can extract the dict information from the response and we can calculate the final score.
> > **According to your suggestion, we add this prompt setting to Appendix D  (L894-L909) of the final version of paper.**
> >
> > [1] Long context transfer from language to vision. arXiv 2024.06
> >
> > [2] Egoschema: A diagnostic benchmark for very long-form video language understanding. NeurIPS 2024
> >
> > [3] A proposed system and its control processes. _The Psychology of Learning
> > and Motivation_ 1968
> >
> > [4] Youtube-8m: A large-scale video classification benchmark. arXiv 2016.09
> >
> > [5] Video-llava: Learning united visual representation by alignment before projection. EMNLP 2024
> >
> > [6] The llama 3 herd of models. arXiv 2024.07

---

> > > ### Comment · Reviewer_9hDR · 2024-12-01
> > > **concerns addressed**
> > >
> > > All my concerns got address. Thank you.

---

> > > > ### Author Response · Authors · 2024-12-01
> > > >
> > > > Thank you for your prompt feedback and for acknowledging that we have addressed your concerns.
> > > >
> > > > In response to your suggestions, we have made several improvements:
> > > >
> > > > *  Section 3 (Lines 254-267): We have enhanced the description concerning the **motivation behind our component design** to provide clearer insights into our methodology.
> > > >
> > > > *  Figures 1, 4, and 5: We addressed the **icon and color issues** to eliminate any visual confusion and aid in better understanding of our structural framework.
> > > >
> > > > * Appendix D (Lines 894-909): We introduced a **detailed judgment process** to ensure that other researchers can accurately reproduce our results.
> > > >
> > > > * Section 4.4, "Design of Long Memory Tree" (Lines 472-473): We **highlighted the vision feature compression process** to elucidate how it influences the effectiveness of input sequence length, aiding in a comprehensive grasp of its operational impact.
> > > >
> > > > We are committed to continual improvement of our manuscript. Although the submission deadline has passed, we will keep improving our work to enhance its quality. **We hope these efforts and revisions meet your expectations and kindly request you to consider raising your rating based on these enhancements. Thank you once again for your constructive feedback.**

---

### Official Review · Reviewer_Lmv2 · 2024-11-10

**Soundness:** 3
**Presentation:** 3
**Contribution:** 4
**Rating:** 8
**Confidence:** 4

**Summary:**

This paper introduces StreamChat, a training-free framework for streaming video reasoning and conversational interaction, and StreamBench, a comprehensive benchmark for evaluating streaming video understanding across various media and interactive scenarios. Extensive evaluations on StreamBench and other public benchmarks show that StreamChat significantly outperforms existing state-of-the-art models in accuracy and response times, highlighting its effectiveness for streaming video understanding.

**Strengths:**

Nice paper. This work is comprehensive, provides significant contributions, and is well-written.

The biggest contribution is StreamBench. Previously, there was no suitable benchmark for streaming video understanding. StreamBench is a high-quality addition.

Furthermore, StreamChat is also very interesting, demonstrating that an agent-like system can also achieve state-of-the-art results in streaming video understanding.

**Weaknesses:**

Overall, the paper is satisfactory, but I have some concerns regarding StreamChat:

- What motivated the selection of LongVA as a foundation? What are the essential capabilities required for other Video-LLMs to serve as the base model?

- Would fine-tuning LongVA with some streaming vision-language data be better than the current system?

- Is it feasible to integrate the proposed hierarchical memory storage into a trainable framework?

**Questions:**

Please refer to the part of weaknesses.

---

> ### Author Response · Authors · 2024-11-23
> **Official Comment by Authors**
>
> **Thank you for your insightful feedback and appreciate your recognition of the novelty of our proposed StreamBench and StreamChat architecture.**
>
> **Our responses to your concerns are detailed below:**
>
> **Q1:Motivations of foundation model selection.**
>
> **A1:Thank you for this insightful question regarding our choice of foundation model.** We selected LongVA [1] for two primary reasons:
>
> **First, LongVA demonstrated state-of-the-art performance among video-language models at the commencement of our research.**  Its foundation in the powerful Qwen2 [2] LLM allows it to handle complex questions and leverage extensive pre-trained knowledge.
>
> **Second, the availability of open-source weights for LongVA facilitated** rapid experimentation and adaptation to various applications, which was crucial for our research.
>
> To dispel your doubts, we are also tried to integrate more models. We replaced VILA as our baseline models and completed the test on StreamBench. The main reason why we select VILA is **its open-source code makes it easy for us to supply our system with it.**
>
> Here is the parameter setting.
>
> | $t$ | $L$ | $g$ | $C$ |
> | --- | --- | --- | --- |
> | 0.35 | 20 | 10 | 2 |
>
> In order to make VILA stable for inference, we reduce $C$(clustering goal) to 2 to reduce the computational budget (23GB).
>
> | Method | FPS | Fr. | Sco. | Acc. | Coh. | RPD |
> | ---- | ---- | ---- | ---- | ---- | ---- | ---- |
> | VILA1.5            | -- |  8 | 3.10 | 57.1 | 2.20 |  --  |
> | StreamChat-VILA1.5     | **22** |  -- | **3.12** | **57.9** | **2.26** |  **0.83**  |
>
> | ~ | OS | LM | SM | CI | KG | SF |
> | --- | --- | --- | --- | --- | --- | --- |
> | Method  | Sco./Acc. | Sco./Acc. | Sco./Acc. | Sco./Acc. | Sco./Acc. | Sco./Acc. |
> | VILA1.5               | **2.33/36.1** | 2.54/44.3 | **2.87/50.8** | 3.59/68.3 | **3.97/78.6** | 3.38/65.5 |
> | StreamChat-VILA1.5        | 2.21/34.4 | **2.74/46.5** | 2.77/49.6 | **3.64/71.1** | 3.83/76.4 | **3.39/65.7** |
>
> According to our experiment, we found our system can further improve VILA model's performance in StreamBechm, especially in Long-term Memory (LM) and Conversational Interaction (CI) tasks.
> There is a certain fluctuation in the performance of the model on OS(Object Search) , KG(Knowledge-based) and SM(Short-term Memory) tasks.
> We believe there are two reasons.
>
> **First, particularly sensitive to prompts.** Our system needs to retrieve historical information from the dialogue memory as new context, and these new contexts need to form new prompts with the original questions.
>
> **Second, hard to accommodate long context.** The information retrieved from the memory includes not only text, but also compressed visual tokens. As shown in **Figure 6,** the prompt composed of these information is longer than the original question.
>
> **We acknowledge and appreciate your suggestion.**
> As we discovered in this experiment, we are trying to further improve our system in the following aspects:
>
> **1. Specilized prompts.** We are continuing to improve more specialized video-llms prompts. Different video-llm may need to formulate specialized prompts based on the distribution of training data, the choice of language model, and other conditions to achieve better results on our system.
>
> **2. Adapt more models.** Not all video-llms are trained with long context data. We are continuing to adapt our system to more video-llms or multi-modal llms, even though they can process limited video frames.

---

> > ### Author Response · Authors · 2024-11-23
> > **Official Comment by Authors**
> >
> > **Q2: Characteristics that a basic model needs to have.**
> >
> > **A2: Thank you for your attention to the characteristics of the base model.**
> >
> > Our research indicates that a suitable model should possess the following key attributes:
> >
> > **1. Long-Form Video Understanding:**  Effective processing of long videos is crucial. While we utilize K-Means for feature compression, the information retrieved by our memory mechanism remains relatively long, requiring a model capable of handling extended sequences.
> >
> > **2. Robustness to Prompt Variations:**  For accurate and coherent multi-turn conversations, the model must be robust to changes in prompt wording.  This robustness is essential to prevent inconsistencies or hallucinations in the model's output when prompts are adjusted to incorporate information from the memory mechanism.
> >
> > By integrating LongVA with our proposed system, we successfully extend its capabilities to encompass streaming video processing and multi-turn conversations while preserving these critical characteristics.
> >
> > **Inspired by your advice, we conducted experiments using the open source model LongVILA.**
> > Here is the parameter setting that we utilized.
> >
> > | $t$ | $L$ | $g$ | $C$ |
> > | --- | --- | --- | --- |
> > | 0.35 | 35 | 10 | 5 |
> >
> > | Method | FPS | Fr. | Sco. | Acc. | Coh. | RPD |
> > | ---- | ---- | ---- | ---- | ---- | ---- | ---- |
> > | LongVILA           | -- |  8 | 3.10 | 50.1 | 2.20 |  --  |
> > | **StreamChat-LongVILA**     | **22** |  -- | **3.15** | **52.3** | **1.98** |  **1.04**  |
> >
> > | ~ | OS | LM | SM | CI | KG | SF |
> > | --- | --- | --- | --- | --- | --- | --- |
> > | Method  | Sco./Acc. | Sco./Acc. | Sco./Acc. | Sco./Acc. | Sco./Acc. | Sco./Acc. |
> > | LongVILA        | 2.33/36.1 | 2.54/44.3 | 2.87/50.8 | 3.59/68.3 | 3.97/78.6 | 3.38/65.5 |
> > | **StreamChat-LongVILA** | **2.38/38.0** | **2.56/45.8** | **2.94/51.2** | **3.60/69.1** | **3.98/78.9** | **3.42/66.7** |
> >
> > According to the experiment result, we found that for LongVILA, our system can further improve 2.2\% accuracy.
> > **We will continue to upgrade more video understanding models to further improve the versatility of our method.**
> >
> > **Q3: Would fine-tuning LongVA with some streaming vision-language data be better?**
> >
> > **A3: Thank you for this insightful question about fine-tuning LongVA.**  Our research focuses on improving performance under resource constraints, and a training-free approach is crucial for achieving this with minimal resource consumption. While existing methods like VideoLLM-online [3], Flash-VStream [4], and VideoStreaming [4] explore fine-tuning with self-constructed instruction data, they face certain limitations:
> >
> > **1. Limited Generalization:**  Methods like VideoLLM-online, while leveraging LLM-generated training data, often achieve suboptimal performance on general video understanding benchmarks.  We hypothesize that fine-tuning large pre-trained LLMs (trained on millions or even billions of data points) can negatively impact their generalization ability , which is essential for adapting to diverse scenarios.
> >
> > **2. Reduced Processing Speed:**  The inherent structure of existing fine-tuning methods has limited video processing speed (with the fastest achieving 5 FPS), a significant drawback for real-time applications.
> >
> > While we acknowledge the potential of fine-tuning, as demonstrated by the partial success of VideoLLM-online, these limitations motivated our exploration of a training-free approach, which has shown promising initial results.
> > **Our future work will focus on identifying limitations of our training-free approach and further investigating trainable paradigms that address these challenges.**
> >
> > **First, exploring high-quality streaming video instruct-tuning data.** We believe the quality of data is improtant to improve the generalization of video-llms. More video scenes and more training tasks should be included in the instruct-tuning data. What's more, the data filtering is crucial. Filtering out toxic text and data with low information content can effectively improve the quality of training data. We have accumulated a lot of experience in benchmark data piepline as introduced in Appendix A and are confident that it can be transferred to production of training data.
> >
> > **Second, developing more dynamic model structure.** In streaming video scenarios, inference efficiency can also be important. We are exploring more dynamic model structure to overcome the tight budget for time and space resource consumption.

---

> > > ### Author Response · Authors · 2024-11-23
> > > **Official Comment by Authors**
> > >
> > > **Q4: Integrate the proposed hierarchical memory storage into a trainable framework.**
> > >
> > > **A4:** While integrating our hierarchical memory storage into a trainable framework is theoretically feasible, as demonstrated by prior work on trainable multi-round conversational models for streaming video [5], resource constraints (both computational and time) within the rebuttal period prevent us from conducting the necessary experiments.  Exploring trainable approaches and leveraging larger, high-quality video instruction-tuning datasets are key priorities for our future work.
> > >
> > > [1] Long context transfer from language to vision. arXiv 2024.06
> > >
> > > [2] Qwen2 technical report. arXiv 2024.07
> > >
> > > [3] Videollm-online: Online video largelanguage model for streaming video. CVPR 2024
> > >
> > > [4] Flash-vstream: Memory-based real-time understanding for long video streams. arXiv 2024.06
> > >
> > > [5] Streaming long video understanding with large language models. arXiv 2024.05
> > >
> > > [6] Longvila: Scaling long-context visual language models for long videos. arXiv 2024.08

---

> > > > ### Comment · Reviewer_Lmv2 · 2024-11-25
> > > >
> > > > Thank you so much for the rebuttal. All of my concerns have been addressed.
> > > >
> > > > Regarding the data size issues raised by other reviewers, I think the current data scale is sufficient (a bit more might be beneficial, e.g., +1K). Note that VideoMME also has only 2.7K QA pairs. Too many evaluation samples cost too much time. LMMs have so many benchmarks that need to be evaluated.

---

> > > > > ### Author Response · Authors · 2024-11-25
> > > > > **Official Comment by Authors**
> > > > >
> > > > > Thank you so much for your prompt feedback and recognition of our responses. If you still have any questions or further suggestions, please feel free to ask. We welcome any opportunities to improve our paper.
> > > > >
> > > > > Your effort and thoughtful suggestions have helped us to improve this manuscript, and we will continue to improve the benchmark in terms of quantity and quality.

---

### Author Response · Authors · 2024-11-23
**General response to all reviewers**

We sincerely thank all reviewers for their thoughtful feedback. **We appreciate the positive comments on the significance of our benchmark and the compelling results of our streaming video understanding system.**  We have carefully addressed all comments and revised the manuscript accordingly. Changes are highlighted in blue.  Key updates include:

**1. Clarity and Presentation:**  Based on the feedback from reviewers **Lmv2**,**9hDR** and **qLQa**, we clarified the selection and utilization of foundation models (main text and appendix).  Section 3 has been reorganized to better explain the motivation behind our memory component design.  Figures 1, 4, and 5 have been revised with distinguishing icons and improved structure for enhanced clarity.

**2. Additional Results:** As suggested by reviewers **Bg29** and **yVyo**, we expanded the evaluation in Tables 4 and 5 to include results for both closed-source models (GPT-4o and GPT-4o-mini) and open-source models (MiniCPM-V 2.6, InternLM-XComposer 2.5, VILA, and InternVL2), as well as human performance, providing a more comprehensive comparison.

**3. Expanded Model Evaluation:**  In response to reviewer **yVyo's** suggestion, we have evaluated additional open-source models on StreamBench and included their results in Tables 4 and 5.

**4. Data expansion plan.** According to the suggestions from reviewers **Bg29** and **yVyo**, we have provided our benchmak data expansion plan in the appendix, and we are continuing improve the quality of our benchmark from multiple aspects.

We believe these revisions significantly strengthen the manuscript. Please do not hesitate to leave comments with any further questions. We appreciate your continued feedback.

Best regards,

The authors

---

### Meta-Review · Area_Chair_HzyV · 2024-12-21

**Metareview:**

The authors propose a new benchmark and a model for question-answering and multi-turn dialog in videos in a streaming setting. This is a relevant and timely problem, and it is a natural use-case for processing long videos.

Reviewers largely appreciated the dataset and benchmark contribution of this work. As it probes an underexplored area of video understanding research, it may have a significant contribution on the community. On the modelling side, reviewers had some concerns that the approach is not very well motivated, that other alternatives could have been considered and ablated, and that it involved too much systems engineering. Authors are also encouraged to discuss prior [streaming architectures](https://openaccess.thecvf.com/content/CVPR2024/papers/Zhou_Streaming_Dense_Video_Captioning_CVPR_2024_paper.pdf) in the revision.

Whilst Reviewer yVyo was overall negative on the paper, the AC believes that the authors have sufficiently addressed the reviewers concerns (ie comparisons with prior work, dataset size) in the rebuttal and in the revision of the paper.

Therefore, the final decision is to accept the paper.

**Additional Comments On Reviewer Discussion:**

Refer to above. Whilst Reviewer yVyo was overall negative on the paper, the AC believes that the authors have sufficiently addressed the reviewers concerns (ie comparisons with prior work, dataset size) in the rebuttal and in the revision of the paper.

---

### Decision · Program_Chairs · 2025-01-22

Accept (Poster)